# Impact Deposition Behavior of Al/B_4_C Cold-Sprayed Composite Coatings: Understanding the Role of Porosity on Particle Retention

**DOI:** 10.3390/ma16062525

**Published:** 2023-03-22

**Authors:** Hannaneh Manafi Farid, André McDonald, James David Hogan

**Affiliations:** Department of Mechanical Engineering, University of Alberta, Edmonton, AB T6G 2R3, Canada

**Keywords:** aluminum matrix, boron carbide, cold spray, cold-sprayed composite coatings, particle reinforced metal matrix composites, porosity

## Abstract

This study explores the role of porosity in the impact deposition of a ceramic-reinforced metal-matrix (i.e., Al/B${}_{4}$C) composite coating fabricated via cold spraying. The Johnson–Holmquist–Beissel constitutive law and the modified Gurson–Tvergaard–Needleman model were used to describe the high strain-rate behavior of the boron carbide and the aluminum metal matrix during impact deposition, respectively. Within a finite element model framework, the Arbitrary Lagrangian–Eulerian technique is implemented to explore the roles of reinforcement particle size and velocity, and pore size and depth in particle retention by examining the post-impact crater morphology, penetration depth, and localized plastic deformation of the aluminum substrate. Results reveal that some degree of matrix porosity may improve particle retention. In particular, porosity near the surface facilitates particle retention at lower impact velocities, while kinetic energy dominates particle retention at higher deposition velocities. Altogether, these results provide insights into the effect of deposition variables (i.e., particle size, impact velocity, pore size, and pore depth) on particle retention that improves coating quality.

## 1. Introduction

Particle-reinforced metal matrix composite (PRMMC) coatings (e.g., Al/B${}_{4}$C [1], Al/SiC [2], Al/Al${}_{2}$O${}_{3}$ [3]) have been widely employed in a variety of applications (e.g., aerospace [4], automotive [1,5], fuel storage [6,7,8], and transportation [9]) because of their favorable tribological properties [6,10,11,12], high hardness and stiffness [10], and fatigue resistance [13]. Typical manufacturing methods for PRMMCs include friction stir [14], squeeze casting [15], stir casting [16], powder compaction [17], and thermal spraying [18]. Among these techniques, the cold spray method [19] was recently adopted because of its favorable attributes: (1) the low temperature of the process ensures no phase change in the material [19], (2) materials with different thermal properties (e.g., ceramics and metals with very different melting points) and morphology can be mixed in the feedstock during the deposition process towards fabricating composite coatings [19], (3) cold spraying does not produce oxidation in the deposited coating [20], and (4) this technique generates lower residual stresses in the final coating when compared to other thermal spray techniques [21]. These features make the cold spray method a unique technique to manufacture composite coatings in order to reduce material consumption and tailor the physical, mechanical, and tribological properties by blending dissimilar materials to provide commercial products for a variety of industrial applications including [21] to repair magnesium parts in aerospace [22], manufacture electro- or thermo-conductive coatings for power electronic circuit boards [23], design of orthopedic devices [22] in biomedical implants [24]. Moreover, cold-sprayed composite coatings are highly appealing because they do not undergo alloying, phase transformation, or thermite reactions during fabrication processes [25].

During the cold spray process, porosity is an indicator of quality because uncontrolled porosity can result in friable structures and, subsequently, poor mechanical properties [26,27]. In cold spraying, deposition parameters (e.g., temperature and velocity of the gas, standoff distance, and angle of spraying [21]) and particle morphology (size, shape, type of particles) significantly affect the porosity level in the final coating [28]. To date, limited studies have focused on exploring the effect of porosity on the mechanical properties and structural integrity of Al/B${}_{4}$C coatings [27,29,30,31,32]. In the literature, Zhao et al. [33] systematically investigated the effect of B${}_{4}$C and Al feedstock particle size on the weight percentage of the B${}_{4}$C in the cold-sprayed Al/B${}_{4}$C coating, and they found the optimal particle size for Al and B${}_{4}$C was 15 μm to maximize the volume fraction of the B${}_{4}$C particles (≈30%), and to achieve maximum deposition efficiency. In the recent study by Zhao et al. [34], they examined the effect of the Al (metal matrix material) particle size on B${}_{4}$C retention and the tribological properties of an Al/B${}_{4}$C cold sprayed composite coating. They found that the smaller Al size facilitated the B${}_{4}$C retention because of the grain refinement of the coatings, which also improved its wear resistance. In another study, Shikalov et al. [28] studied the tribological behavior of Al/B${}_{4}$C coatings for different B${}_{4}$C powder sizes (17 and 75 μm) and volume percentages, and they found that the higher hardness was attributed to the finer particles, and the B${}_{4}$C size had no effect on the adhesion strength. Moreover, other studies [12,21] have shown that hard ceramic particles (e.g., B${}_{4}$C) fragment more after deposition into softer materials (e.g., Al), resulting in interfacial gaps between the particles and substrate, and subsequently, increasing the porosity level. Building on these studies, our efforts in this study aimed at better understanding the role of the matrix porosity (the size and location of pores), particle sizes, and impact velocities on the resulting impact deposition and, by extension, the quality of Al/B${}_{4}$C coatings (i.e., particle retention [12,33]).

To study impact deposition processes in cold spray manufacturing, numerical studies can be used to unravel the effects of deposition parameters (e.g., particle size and shape and impact velocity) by realizing different deposition configurations. Various finite element approaches (e.g., coupled Eulerian–Lagrangian (CEL) [35], arbitrary Lagrangian–Eulerian (ALE) [36,37], and Lagrangian [38]), and material models (e.g., Johnson–Cook [39] and Mie–Gruneisen [40]) have been employed to model impact deposition responses (e.g., bonding strength [41], retention [42], and rebounding times [43]). Through simulations, it has been shown that the increase in particle impact velocity increases the temperature and plastic deformation of the contact area, which has been shown to have an influence on particle retention and final coating strength [44]. For example, Chakrabarty et al. [45,46] numerically studied deposition and retention of a single ceramic particle on a metallic substrate at different particle densities, velocities, and impact angles, and found that the oblique spray angle, higher density, and velocity of the depositing particles resulted in increasing the jetting region which strengthened the particle retention on the substrate. In another study, Elkin et al. [43] numerically studied the role of the surface roughness on the retention of irregular-shaped SiC particles by using the Johnson–Cook [39] model and CEL technique and found that irregular particle shapes resulted in less porosity and better retention when compared to spherical particles [47]. In these studies, the material models for the ceramic particles were simple elastic models and so more physically-relevant ceramic models are needed to better understand the fracture and fragmentation [2,11] of the particles upon impact into the matrix, which will improve our fundamental understanding of the impact deposition processes. This study will address this gap by implementing the Johnson–Holmquist–Beissel model [48], with considerations for the role of matrix porosity on ceramic particle retention.

In this paper, for the first time in the literature on metal–ceramic composites, we investigate the role of particle size and velocity, and matrix porosity on key particle retention parameters (i.e., depth of penetration, crater morphology, matrix plastic strain, and particle damage) during the deposition of a boron carbide ceramic particle into an aluminum matrix with a pore using the finite element approach. The Johnson–Holmquist–Beissel (JHB) [48] and the Gurson–Tvergaard–Needleman (GTN) [49] material models are used for the ceramic particle and aluminum substrate, respectively. We will investigate the effect of particle diameter, impact velocity, pore diameter, and pore location on the particle penetration depth, crater roughness, and plastic deformation across the contact surfaces in the substrate. Altogether, the results generated in this study will provide insights into the role of porosity during impact deposition behavior of Al/B${}_{4}$C composites (e.g., matrix strain, roughness, and penetration depth) towards better informing improved cold spray deposition parameters and, eventually, material optimization.

## 2. Methodology and Model Configurations

To better understand the three-dimensional impact deposition behavior of the Al/B${}_{4}$C coating, and the effect of variables for retention of a B${}_{4}$C particle on an Al matrix with a pore, a numerical explicit finite element simulation using Abaqus 2020 (Dassault Systems Simulia Corp., Johnston, RI, USA) software is employed. In this study, the ceramic particle and metallic matrix are treated as brittle and ductile materials, respectively, which are subjected to high-strain-rate impact loadings, and appropriate material models are used to analyze their true behavior. Therefore, the failure behavior of the ductile aluminum substrate is modeled using the Gurson–Tvergaard–Needleman (GTN) model because of its ability to account for the plastic deformation and void growth mechanism [50]. The Johnson–Holmquist–Beissel (JHB) model is used to describe the elastic-plastic deformation of the ceramic because it can reasonably capture the strain-rate-dependent mechanical response of the brittle boron carbide under impact loadings [51]. The following subsections lay out the major constitutive equations for the material models, the finite element framework of the Arbitrary Lagrangian Eulerian technique (ALE) [52], and model configurations used in this study.

### 2.1. The Gurson–Tvergaard–Needlman Model

The Gurson–Tvergaard–Needlman (GTN) model is a well-known elastic-plastic micro-mechanical model that accounts for the ductile damage accumulation in terms of void nucleation, growth, and coalescence [53]. The GTN model is based on continuum damage mechanics and introduces a failure criterion [49]:(1)$$\phi \left(\sigma ,f\right)={\left(\frac{\sigma_{eq}}{\sigma_{y}}\right)}^{2}+2q_{1}f^{*}\cosh\left(\frac{3q_{2}\sigma_{m}}{2\sigma_{y}}\right)-\left(1+q_{3}f^{*2}\right)=0,$$
where $q_{1,2,3}$ are constitutive parameters, $\sigma_{m}$ is the hydrostatic or mean normal stress ($\sigma_{m}=\left(\sigma_{11}+\sigma_{22}+\sigma_{33}\right)/3$), $\sigma_{eq}$ is the Von Mises equivalent stress $\left(\sqrt{3S_{ij}S_{ij}/2}\right)$, $S_{ij}$ is the deviatoric stress tensor, $\sigma_{y}$ is the yield stress, and $f^{*}$ represents the modified damage parameter and porosity, which is a function of the volume fraction, *f*, and defined as:(2)$$f^{*}=\left\{\begin{array}{cc} f & f\leq f_{c}\\ f_{c}+\frac{f_{u}^{*}-f_{c}}{f_{F}-f_{c}}\left(f-f_{c}\right) & f_{c}\leq f\leq f_{F},\\ f_{u}^{*} & f\geq f_{F} \end{array}\right.$$
where $\frac{f_{u}^{*}-f_{c}}{f_{F}-f_{c}}\left(f-f_{c}\right)$ represents the final phase of ductile failure, $f_{c}$ is the critical void volume fraction, $f_{u}^{*}=\frac{q_{1}+\sqrt{q_{1}^{2}-q_{3}}}{q_{3}}$ is the ultimate damage parameter, and $f_{F}$ is the final void volume fraction. The rate of the void volume fraction, $df=df_{\mathrm{nucleation}}+df_{\mathrm{growth}}$, is an addition of nucleation ($df_{\mathrm{nucleation}}$) and the void growth ($df_{\mathrm{growth}}$), where:(3)$$df_{\mathrm{nucleation}}=A_{n}d\varepsilon_{eq}^{p},$$
with
(4)$$A_{n}=\left\{\begin{array}{cc} \frac{f_{N}}{S_{N}\sqrt{2\pi }}e^{-0.5{\left(\frac{\varepsilon^{p}-\varepsilon_{N}}{S_{N}}\right)}^{2}} & \;\mathrm{if}\;\sigma_{m}\geq 0\;\\ 0 & \;\mathrm{if}\;\sigma_{m}<0 \end{array}\right.$$

Here, $A_{n}$ is a function of the void nucleation ($f_{N}$), void nucleation generated strain ($\varepsilon_{N}$), standard deviation of the void nucleation distribution ($S_{N}$), and plastic strain ($\varepsilon^{p}$). Lastly, the void fraction rate due to the void growth is:(5)$$df_{\mathrm{growth}}=\left(1-f\right)d\varepsilon_{ii}^{p},$$
where $\varepsilon_{ii}^{p}$ is the plastic hydrostatic strain. In this paper, parameters for the model are populated from the previous work by the authors [54,55], and the literature [56], and these parameters are presented later in Section 2.3.

### 2.2. The Johnson–Holmquist–Beissel Model

The JHB model is a phenomenological model that describes the failure behavior of brittle materials subjected to large strain, high strain rate, and high pressure [48]. In this study, the JHB model is applied to describe the mechanical behavior of boron carbide (B${}_{4}$C) particles impacting an Al substrate. Previous studies demonstrated that B${}_{4}$C shows a sudden loss in strength under high pressure after the Hugoniot Elastic Limit (HEL) strength is reached [57], for which this behavior can be well captured by the JHB model [58] because it is more physically relevant for our study. In general, this model consists of three main curves [58]: (1) a strength curve for both intact and damaged ceramics, (2) a damage function describing material failure, and (3) a pressure vs. volumetric strain relationship for bulking and phase change. Each of these components are subsequently described.

First, the strength model (von Misses equivalent stress versus pressure) consists of two curves for intact and failed materials. The von Mises equivalent stress, σ, depends on the pressure, *P*, the dimensionless equivalent strain rate, ${\dot{\varepsilon }}^{*}$, where ${\dot{\varepsilon }}^{*}=\dot{\varepsilon }/\dot{\varepsilon_{0}}$ and ${\dot{\varepsilon }}_{0}=1.0$ s${}^{-1}$, and the damage, *D*. D=0, 0<D<1, and D>1, represent the intact, partially damaged, and fully damaged or failed materials, respectively. The strength model for the intact material (D<1) is defined as:(6)$$\sigma =\left\{\begin{array}{cc} \sigma_{i}\left(P+T\right)/\left(P_{i}+T\right) & -T<P<P_{i}\\ \sigma_{i}+\left(\sigma_{max}-\sigma_{i}\right)\left\{1.0-exp\left[-\alpha_{i}\left(P-P_{i}\right)\right]\right\} & P_{i}<P, \end{array}\right.$$
where $\alpha_{i}=\sigma_{i}/\left[\left(\sigma_{max}-\sigma_{i}\right)\left(P_{i}+T\right)\right]$. For failed material (D=1), the strength model is:(7)$$\sigma =\left\{\begin{array}{cc} (\sigma_{f}/P_{f})P & 0<P<P_{f}\\ \sigma_{f}+\left(\sigma_{max}-\sigma_{f}\right)\left\{1.0-exp\left[-\alpha_{f}\left(P-P_{f}\right)\right]\right\} & P_{f}<P \end{array}\right.$$
where $\alpha_{f}=\sigma_{f}/\left[\left(\sigma_{max}-\sigma_{f}\right)\left(P_{f}+T\right)\right]$. *T*, $\sigma_{i}$, $\sigma_{f}$, $P_{i}$, and $P_{f}$ represent the tensile pressure, minimum nonlinear stress of the intact and failed material, and the corresponding pressure at the minimum nonlinear stress of the intact and failed material, respectively. The strain-rate dependent strength for ${\dot{\varepsilon }}^{*}>1$ is:(8)$$\sigma =\sigma_{0}\left(1.0+C\ln{\dot{\varepsilon }}^{*}\right),$$
where $\sigma_{0}$ is the corresponding strength at ${\dot{\varepsilon }}^{*}=1$ obtained from Equation (6) or Equation (7), and *C* is a dimensionless strain rate constant.

Next, the damage model to describe the material failure is defined as:(9)$$D=\Sigma (\Delta \varepsilon_{p}/\varepsilon_{p}^{f}),$$
where $\Delta \varepsilon_{p}$ is the increment equivalent plastic strain and $\varepsilon_{p}^{f}$ is the constant plastic strain defined as $\varepsilon_{p}^{f}=D_{1}{\left(P^{*}+T^{*}\right)}^{n}$, where $P^{*}=P/\sigma_{max}$, $T^{*}=T/\sigma_{max}$, and dimensionless $D_{1}$ and *n* are constants.

Lastly, the hydrostatic pressure model is based on the volumetric strain, $\mu =\frac{V_{0}}{V}-1=\frac{\rho }{\rho_{0}}-1$. *V*, ρ, $V_{0}$, and $\rho_{0}$ are current volume and density, and initial volume and density, respectively. The pressure model with a phase change and before damage (D< 1) is defined as:(10)$$P=\left\{\begin{array}{cc} K_{1}\mu +K_{2}\mu^{2}+K_{3}\mu^{3} & 0<\mu <\mu_{1}\\ \; & 0<P<P_{1}\\ \left(\frac{P_{2}-P_{1}}{\mu_{2}-\mu_{1}}\right)\mu +\frac{P_{1}\mu_{2}-P_{2}\mu_{1}}{\mu_{2}-\mu_{1}} & \mu_{1}<\mu <\mu_{2}\\ \; & P_{1}<P<P_{2}\\ {\bar{K}}_{1}\bar{\mu }+{\bar{K}}_{2}{\bar{\mu }}^{2}+{\bar{K}}_{3}{\bar{\mu }}^{3} & \mu >\mu_{2}\\ \; & P>P_{2} \end{array}\right.$$
where $K_{1}$ (bulk modulus), $K_{2}$, $K_{3}$, ${\bar{K}}_{1}$, ${\bar{K}}_{2}$, ${\bar{K}}_{3}$, and $\mu_{0}$ are constants. $P_{1}$ is the maximum pressure at phase 1 at $\mu_{1}$ and $P_{2}$ is the minimum pressure at the beginning of phase 2 at $\mu_{2}$. The transition pressure from phase 1 to phase 2 is a linear model. After the material fails (*D* = 1), bulking occurs, and the change in pressure is added to the Equation (10). For instance, the pressure model for the failed materials (D=1) and μ>0 is:(11)$$P=K_{1}\mu +K_{2}\mu^{2}+K_{3}\mu^{3}+\Delta P,$$
where ΔP is the pressure increment showing the material bulking after failure, and it is obtained considering the change in the internal elastic energy:(12)$$\Delta p=-K_{1}\mu_{f}+\sqrt{{\left(K_{1}\mu_{f}\right)}^{2}+2\beta K_{1}\Delta U},$$
where ΔU is the internal elastic energy loss when failure occurs, $\mu_{f}$ is the volumetric compression at failure, and β (0≤β≤1) is the fraction of the internal energy loss converted to potential hydrostatic energy.

### 2.3. Impact Deposition Model Configurations

In this study, a three-dimensional (3D) model of a B${}_{4}$C particle impacting on an Al substrate is modeled in an Abaqus/Explicit framework to systematically study the cold spray impact deposition process of ceramic/metal composites with a pore. The B${}_{4}$C particle is regarded as deformable with elastic-plastic behavior taken into account, and this is in contrast to the previous models where ceramic particles are assumed to be elastic using isotropic elastic models [43,46,59,60]. Table 1 summarizes the JHB constants for the B${}_{4}$C particle, and these constants are extracted from the previous study by Johnson and Holmquist [61]. In addition, Table 2 also summarizes the GTN model constants, which are taken from Sayahlitifi et al. [55] and modified based on the literature [54,56] to describe the ductile failure of the Al substrate with void growth considered.

Figure 1 shows the model geometry used for the numerical impact simulations. This model was inspired by the 3D models of a cold sprayed single particle in the literature [44,46,62,63] and the high velocity impact modeling example in Abaqus [64]. While the B${}_{4}$C particle has an irregular morphology [65], which results in better retention in the Al matrix and higher reinforcement contents, in this study, the particle shape is assumed to be spherical to simplify the simulation. Ceramic particles with spherical shapes may have a lower reinforcement fraction [66]. However, they are more likely to increase the in situ hammering effect, which enhances grain refinement and structure density [67]. Owing to the axi-symmetry of the geometries (a cylindrical substrate and a spherical particle) and loading (perpendicular impact of a particle on a substrate), a slice that is the 1/32 of the entire model following [68] is used to reduce the computational time in this study. In Figure 1, three sets of boundary conditions are applied: (1) along the symmetric axis where all nodes can only move along the Z axis, (2) on all the side surfaces of the particle and substrate where circumferential velocities of all nodes are zero, and (3) at the substrate bottom where all nodes cannot move along the Z-direction. These boundary conditions are consistent with the literature [64].

In this study, the B${}_{4}$C particle diameter varies from 15, 25, to 40 μm (Table 3), and this is guided by the particle size in experimental studies of Zhao et al. [33]. The substrate height ($H_{Substrate}$) and radius ($R_{Substrate}$) are chosen to be 75 μm to avoid any possible wave reflection (i.e., estimated by using the elastic wave velocity equation, *v* $=\sqrt{\left(E/\rho \right)}$). In this study, simulations are performed over the first 24 ns of impact, which is sufficient to allow for observed behavior to be completed; this also corresponds to before when the elastic wave is reflected from the rear of the substrate to return to the impact zone [69]). In addition, the pore shape is assumed to be spherical to simplify the simulation, and the pore diameters are selected to be 1, 2, 3, and 4 μm. Pore depths are 0.1$D_{Particle}$–0.5$D_{Particle}$ based on the observation made in microscopic images of Al/B${}_{4}$C composites from the literature [2,11,12,28,33,70]. Lastly, Figure 1 also demonstrates the meshed particle and substrate with refined areas near the contact region. The mesh size is chosen to be at most 1/50 of the particle diameter near the contact area to avoid element distortion [43,44,46]. The friction coefficient is considered to be 0.25 for the contact between particle and substrate, and this is guided by the literature [71]. The interaction between the particle and the substrate is defined using the general contact algorithm, which has been implemented previously in the literature [44,46,72]. For employing the ALE method [70], the frequency is set at ten as a default value [70,73], and the number of remeshing sweeps per increment is set between 5 to 8 for various models in order to avoid errors in analysis [74,75]. The FS parameter, which is used to minimize the error between numerical and experimental results, is set at 1.5, based on Chalmers [76]. The eight-node linear brick element (C3D8R) with a reduced integration technique and default hourglass is selected for both particle and substrate. The total number of elements for particle and substrate is between 63,227 and 155,126 elements in the simulations. This type of mesh has also been used in the literature [45,46,63]. Compute Canada clusters with one node are used to perform the high-powered parallel computing and minimize the computational time, with a mean run duration of approximately 9 +/− 1 h per simulation on one node, depending on velocity, particle size, and depth.

## 3. Results and Discussions

Given this study’s wide range of numerical data, we classify the results and discussion based on the outputs. Specifically, Section 3.1 compares our model with those in similar studies to verify the model. Section 3.2 describes how the particle penetration depth is affected by deposition parameters (e.g., particle size, impact velocity, and pore sizes and depths). Section 3.3 explores the effect of the pore volume change on the plastic strain of the substrate. Section 3.4 investigates the crater morphology following impact. Section 3.5 examines the pore size and depth on the equivalent plastic strain (PEEQ) value. Next, Section 3.6 investigates the effect of impact velocity, particle size, and pore size on the equivalent plastic strain of the contact surface in the substrate. Finally, Section 3.7 explores results on the localized plastic strain over the contact surface in the substrate. In all areas, we focus on the role of porosity and impact deposition variables (e.g., particle size and velocity) on resulting parameters (e.g., penetration depth, crater roughness, and plastic deformation) that are believed to be associated with particle retention [78,79].

### 3.1. Model Evaluation

In this sub-section, the predictive capability and accuracy of the model are illustrated through the selection of the mesh size, the trend of accumulation of the equivalent plastic strain (PEEQ) for both substrate and particle, and the PEEQ magnitude compared to the previous studies [44,46]. In addition, the effect of the particle material type on the plastic strain will be briefly discussed. Prior to presenting these results, it is worth noting that the model developed in this study is challenging to experimentally validate with in situ deposition data given the small scale of the particles and pores [2,11,12,28,33], and high speeds of the impact deposition process [12,28]. Regardless, we attempt to demonstrate how our model aligns with previous published works.

First, Figure 2 compares normalized computational time and the number of elements across different mesh sizes: 0.2, 0.3, 0.4, and 0.5 μm. We use this plot to inform about the tradeoffs between the number of elements and computational time for physically-relevant mesh sizes, as previously done in the literature [44,46,78]. We employ the ALE technique, an adaptive meshing tool, to avoid excessively distorted elements and the analysis stops. In the ALE technique, the mesh exposed to excessive distortion is replaced by a mesh domain, whose nodes are placed in the interior of the mesh domain, reducing the overall distortion of the material. However, the mesh nodes and material points lose their correspondence at each re-meshing time, which causes an error in the final results showing the material behaviors [80]. In this case, the mesh convergence analysis, one of the most effective methods to validate a finite element model, is inaccurate [19]. Therefore, mesh sensitivity analysis is not beneficial to validate this model. Instead, the method that we used here to validate the model compromises between the element numbers (capturing the relevant physics based on the literature [43,44,81]) and computational time, which has been used in the literature to compare the different numerical methods (e.g., CEL, ALE, and Lagrangian) to show the efficiency and accuracy of the models [44]. It is worth noting that each numerical FE technique can simulate a different aspect of the cold spray deposition process [72]. The ALE technique provides high precision and is mainly used in the literature [70,73] to simulate the build-up process of coatings and multi-particle impacts [82]; considering its advantages, we will also use the ALE technique here for further analysis in the future.

To validate the model, the selected mesh sizes are 0.2, 0.3, 0.4, and 0.5 μm [43,44,81], which are also employed in the cold sprayed model in the literature. These mesh sizes are implemented for a particle diameter of $D_{Particle}$ = 15 μm at the contact areas of the particle and substrate, and near the pore in the substrate. Specifically, these mesh sizes are chosen to be at most 1/50 of the particle diameter (i.e., ≤0.3 μm) guided by the literature [46,78,79] in order to reduce mesh sensitivity of the model. It is observed from Figure 2 that at the mesh size of 0.3 μm (with the normalized computational time of 0.148 and the normalized number of elements of 0.991), the number of elements and the associated computational time are notably lower than mesh sizes of 0.2 μm (with the normalized computational time of 2.163 and the normalized number of elements of 2.431). In contrast, the computational times are comparable between the mesh size of 0.3 and 0.4 μm. In conclusion, a refined mesh size of 0.3 μm is chosen in this study for contact areas to balance the accuracy and computational costs [44,46,78].

Next, Figure 3 shows the PEEQ value over the contact-surface of the substrate and the particle simulated in this study and compared with similar models in the literature [44,45]. We compare this to determine if our implementation produces results of similar magnitudes and trends to those published in the literature concerning the impact deposition of ceramic particles into metal substrates [43,44,46,72,79]. In Figure 3a, the PEEQ value over the Al contact-surface in our Al/B${}_{4}$C coating in the current study and in Al/Al coating reprinted from the literature [44] with a particle diameter of 25 μm and impact velocity of 700 m/s shows that the plastic strain of the Al substrate in this simulation is comparable to previous studies [44]. The solid line in Figure 3a represents the PEEQ value over the Al contact-surface for the Al/B${}_{4}$C cold sprayed coating in this study. For this simulation, the Arbitrary Eulerian–Lagrangian (ALE) method is employed to study the deposition behavior of the particle. The dashed lines represent the PEEQ value over the Al contact-surface in the substrate of Al/Al coating simulated using the Coupled Eulerian Lagrangian (CEL) technique reprinted from the study by Xie et al. [44]. The PEEQ curve trends are similar, and there is no further accumulation of PEEQ after about 30 ns for both coatings (Al/B${}_{4}$C and Al/Al). This may be due to strain hardening caused by the high velocity impact on the substrate [78]. However, particle material types (ceramic, B${}_{4}$C, in Al/B${}_{4}$C and metal, Al, in Al/Al) account for the difference in PEEQ values [44]. The degree of particle ductility influences particle deposition in the cold spray process, since the high velocity impact of softer materials results in additional thermal softening [44], leading to larger plastic deformation, especially at the edges of the deformed particle. Conversely, hard particles (e.g., B${}_{4}$C) do not thermally soften and are likely to spall at the edges as a result of high impact velocity and pressure waves [77]. Therefore, softer or more ductile materials contribute more to mechanical interlocking than harder ones [79]. Figure 3b compares the PEEQ value over the entire B${}_{4}$C particle in the Al/B${}_{4}$C coating (in this study) with the PEEQ value over the entire Copper (Cu) particle in the Cu/Cu coating reprinted from reference [45] and shows consistency of two distinct material models: the JHB model and modified Johnson–Cook (JC) model with strain gradient plasticity (SGP). In these two simulations in Figure 3b, the particle diameter is 41 μm, the impact velocity is 650 m/s, and the numerical techniques are Coupled Eulerian–Lagrangian (CEL) for Cu/Cu and Arbitrary Lagrangian–Eulerian (ALE) for Al/B${}_{4}$C. The trend of the average PEEQ over B${}_{4}$C particle using the JHB model is analogous to that of the average PEEQ over the Cu particle using the modified JC model with SGP, demonstrating the consistency between the modified JC with the SGP effect and the JHB material model. This is attributed to the inclusion of the plastic strain rate effect in these two models, which is more representative of dynamic impact loading cases [83,84,85]. Figure 3b also shows that the B${}_{4}$C deformation rate in the Al/B${}_{4}$C coating is higher than the Cu deformation rate in the Cu/Cu coating, which can be attributed to different particle types and material models. High velocity impacts of harder materials (i.e., B${}_{4}$C here) result in greater plastic deformation, which enhances the retention of particles [44,86]. As a result, the JHB model and other models, including the strain-rate-dependency model, play an important role in understanding the mechanical response of materials during impact deposition.

### 3.2. Effect of Pore Size, Particle Size, and Impact Velocity on Penetration Depth

This sub-section will explore the effect of particle size, impact velocity, and pore size on the particle penetration depth. Here, the deeper penetration depth increases the chance of mechanical interlocking; therefore, this is important to quantify in order to better understand particle retention behaviors during deposition into a substrate [46].

Figure 4 shows the penetration depth of the center of the substrate during deposition of a single B${}_{4}$C particle on an Al substrate within 24 ns of impact in order to explore the effect of impact velocities (Figure 4a), particle sizes (Figure 4b), pore sizes (Figure 4c), and pore-to-particle-size ratio (Figure 4d) on the penetration depth. In Figure 4a with a fixed particle diameter (15 μm) and a range of velocities (500, 600, and 700 m/s) impacting on an intact substrate without a pore, deeper penetration occurs for the higher impact velocities. This is well aligned with the numerical results for different impact velocities in the literature [43] since the higher impact velocities result in greater kinetic energy that is the main driver of particle retention in the cold spray technique [19]. Figure 4b shows results with a fixed impact velocity (500 m/s) and different particle sizes (15, 25, and 40 μm), and results show that deeper penetration occurs for larger particle sizes, as expected, given the greater kinetic energy [19]. For implementation in manufacturing, the deeper particle penetration increases the contact surfaces between particle and substrate, thereby enhancing the chance of mechanical interlocking of the particle and improving particle retention in the substrate [46].

Figure 4c shows the combined effect of both pore sizes (1, 2, 3, and 4 μm) and impacting velocities (500, 600, and 700 m/s) on the penetration depth with a pore embedded at a depth of 0.4$D_{Particle}$ into the substrate. The particle size is taken as $D_{Particle}$= 15 μm. The pore depth of 0.4$D_{Particle}$ is selected based on the literature [12,28] because there is no noticeable change in behavior for penetration depths in the range of 0.4$D_{Particle}$ to 1$D_{Particle}$, while no clear trend is observed for the range of 0.1$D_{Particle}$ to 0.3$D_{Particle}$. In Figure 4c, it is observed that higher impact velocities result in deeper penetrations in all cases with different pore diameters, and this is expected due to higher kinetic energy [19], and numerically demonstrated in the literature [43] that the higher impact velocities of the SiC particle result in deeper penetrations. In addition, the effect of including the pore on the penetration depth at higher velocities (i.e., 600 and 700 m/s) is minor. Specifically, the penetration depth for all considered pore diameters is approximately 4 and 5 μm at impact velocities of 600 and 700 m/s, which are close to the penetration depth for the cases without a pore from Figure 4a. Conversely, at the lower impact velocity of 500 m/s, the inclusion of pores has a noticeable influence on the penetration depth compared to the higher impact velocities. Namely, at $V_{Impact}$ = 500 m/s, the penetration depth for different pore diameters except for a pore with $D_{Pore}$ = 1 μm is generally more than 4 μm, with no obvious trends for the 2 and 3 μm sizes for ranging impact velocities. This is likely related to the complex interplay of particle comminution [12], matrix plastic deformation [87], pore crushing and tamping effect [21], and wave mechanics [77] occurring at these small length scales and short time scales in this single particle impact process. Finally, the particle penetrates nearly twice as much as the case without a pore (see Figure 4a) or the case with a pore diameter of 1 μm (see Figure 4a). Overall, it is believed that the impact velocity has a more significant influence on penetration and is more controllable during cold spray than the pore diameter because of the importance of higher kinetic energy on retention of the particles [19].

Lastly, Figure 4d presents results on the effect of a pore with $D_{Pore}$ = 4 and 8 μm at a depth of 0.4$D_{Particle}$ from the surface with different particle sizes: 15, 25, and 40 μm to show the effect of different pore-to-particle-size ratio on the penetration depth. As before (Figure 4b), the larger particles result in deeper penetrations, and this is believed to be related to greater kinetic energy [19]. In Figure 4d, The penetration depth curves for the pore-to-particle-size ratio of particle diameters of 25 and 40 μm do not plateau at the truncated time of the simulation (i.e., t = 24 ns), indicating that larger particles will penetrate deeper for a longer period. The penetration depth increases from 2 in Figure 4b to 4.2 μm in Figure 4d for the pore-to-particle-size ratio of 0.27 ($\frac{D_{Pore}}{D_{Particle}}=\frac{4}{15}$), increases from 5 to 8 μm for the pore-to-particle-size ratio of 0.16 ($\frac{D_{Pore}}{D_{Particle}}=\frac{4}{25}$), increases from 6.5 to 8 μm for the pore-to-particle-size ratio of 0.1 ($\frac{D_{Pore}}{D_{Particle}}=\frac{4}{40}$), and increases from 6.5 to 10 μm for the pore-to-particle-size ratio of 0.2 ($\frac{D_{Pore}}{D_{Particle}}=\frac{8}{40}$) in Figure 4d. These results show that a higher pore-to-particle-size ratio causes deeper particle penetration (the pore-to-particle-size ratio of 0.27 has a maximum penetration depth increase of approximately $\frac{4.2}{2}$ times). In this study, pore-to-particle-size ratios equal to or less than 0.16 have a limited influence on the penetration depth. To the authors’ best knowledge, no studies have investigated the effect of the pore-to-particle-size ratio on particle retention. However, the SEM images in the literature [2,12,28,33] show larger pore sizes for larger particles and smaller pores for relatively smaller particle diameters. The results suggest that the ratio of the pore to particle size should be considered in manufacturing the porous structure using cold spraying [72].

### 3.3. Effect of Change in Pore Volume on the Equivalent Plastic Strain

This sub-section will explore the effect of the pore volume change on the PEEQ value toward better understanding its effect on particle retention [21]. Figure 5 illustrates the typical time-evolving changes in pore morphology and the values of PEEQ on the contact-surface in the substrate during the impact of a particle with $D_{Particle}$ = 15 μm at $V_{Impact}$ = 500 m/s on a substrate including a pore with $D_{Pore}$ = 4 μm at a depth of 0.3$D_{Particle}$. This figure is shown to better understand the pore volume change during the deposition, and its effect on the PEEQ value, with numerical conditions motivated by the literature [21,88]. Figure 5a shows the average PEEQ value over the contact-surface in the substrate, indicating that the maximum average PEEQ value is 6, which is used in the following figures. Figure 5b–f are the time-resolved still frames corresponding to the different time stamps (5, 10, 15, 20, and 24 ns), and these are chosen to demonstrate the pore morphology, the crater morphology, and the PEEQ value over the Al contact-surface during pore collapsing. At a time of t= 5 and 10 ns (Figure 5b,c), the pore is collapsing as the PEEQ value over the Al contact-surface is increasing, particularly near the crater edges, and jetting begins to happen at t= 10 ns at the crater edges. However, the PEEQ value is lower than the contact-surface average PEEQ value of 6. Figure 5d shows that the pore at time of t= 15 ns is not completely collapsed, and the crater edges demonstrate a higher PEEQ value than the crater center, as well as a larger amount of plastic deformation or jetting. The pore collapses at approximately 17 ns, which is determined by tracking the volume of the pore through time. At *t* = 20 ns in Figure 5e, the pore has already collapsed, and the PEEQ value has increased to 6.9, which is higher than the average PEEQ value of 6 (see Figure 5a). Increasing the PEEQ value without changing the pore morphology improves the particle retention in the Al substrate because the kinetic energy converts to plastic deformation energy at the contact-surface rather than changing the pore morphology, which will be discussed in detail in subsequent figure descriptions. Finally, Figure 5f shows a further increase in PEEQ value across the Al contact surface, particularly near the crater edges, while the crater morphology, specifically at the crater edges, does not change. This plateauing behavior can be related to the saturation in strain hardening at the contact surface of the particle due to the high impact velocity [78]. More specifically, high impact velocities promote the plastic deformation of the matrix, resulting in surface hardening through the tamping effect that leads to the collapse of the interfacial gaps, flaws, and surface porosity, as well as strengthens the bonding at the metal/ceramic interfaces [21,89].

### 3.4. Effect of Pore Size, Particle Size, and Impact Velocity on Crater Morphology

Next, we explore the effect of particle size, impact velocity, and pore size on the crater morphology. Surface morphology and its roughness is an effective parameter for improving particle retention [43,46]. Figure 6 illustrates the side and top views of the impact crater at different configurations regarding substrates, pore sizes, and impact velocities at a fixed particle diameter (i.e., 15 μm) and time (i.e., t = 24 ns). Additionally, the PEEQ vs. time curve for impact velocities of 500 and 700 m/s demonstrates the maximum average PEEQ value over the crater surface used in this figure to determine the distribution of the PEEQ. Figure 6a shows the Al substrate crater morphology without a pore at an impact velocity of 500 m/s. A smooth crater without any discontinuous bump across the crater is observed with a graduate increase in PEEQ value towards the crater center. In addition, the PEEQ value at the center is higher than the averaged PEEQ value, indicating localized plastic deformation and stronger bonding at the crater center, as discussed in the literature [79]. As explained in Figure 4, the penetration depth is lower for the case without a pore compared to the other cases containing a pore, which applies for all sub-figures here. In Figure 6b with $D_{Pore}$ = 1 μm at a depth of 0.3$D_{Pore}$ and $V_{Impact}$ = 500 m/s, the deformation at the crater edges is similar to the case without a pore (Figure 6a), and the crater center displays higher plastic strains, which is in agreement with the literature [79]. In Figure 6c–e, the crater shapes from the side views are more non-uniform and uneven, and the crater edges are distorted. The higher PEEQ value is found at the edges of the crater rather than its center for the cases with complete (Figure 6c or Figure 6d) or partial (Figure 6e) pore collapses. The severe plastic deformation at the crater edges or jetting [77,90] occurs in the cases containing a pore with a diameter greater than 1 μm (i.e., Figure 6d–f) compared to Figure 6a,b. The jetting implies that the particle localized fragmentation occurs at the crater edges and results in material flowing near the crater edges, which has also been shown in the literature [77]. Specific particle fragmentation and material flow behaviors will be further explored in Section 3.7. Figure 6f shows a substrate that includes a pore with a diameter of $D_{Pore}$ = 4 μm at a pore depth of 0.4$D_{Particle}$ at $V_{Impact}$ = 700 m/s, with the saturation of plastic deformation (illustrated as gray color) with PEEQ values of 8.6 being observed across most of the contact surface. Comparing the crater deformation in Figure 6a,f without a pore at $V_{Impact}=$ 500 m/s reveals that the substrate material expands more at the crater edges, the deeper penetration occurs at the crater center, and there is a bump near the middle of the crater, which is attributed to the higher impact velocity and higher kinetic energy generation [77,78]. Altogether, these results are important because they show that, generally, a pore in the substrate significantly contributes to particle deposition. Specifically, the partial or complete pore collapse results in a deeper penetration (see Figure 4) and also leads to the non-uniform crater shapes (see Figure 6) of the contact-surface and excessive distortion of the crater edges (see Figure 5 and Figure 6). The pore effect on the crater morphology and deeper penetration can facilitate improved mechanical interlocking, subsequent particle retention, and finally, increased deposition efficiency of ceramics in the coating [43,46,77].

### 3.5. Effect of Pore Size and Depth on the Time-Evolved Equivalent Plastic Strain

In this sub-section, we explore the effect of pore size and depth on the average PEEQ value over the contact surface in the substrate at an impact velocity of 500 m/s, motivated by deposition conditions from the literature [21]. The PEEQ value indicates the contact-surface’s plastic deformation, which contributes to the localized softening of a thin (few micrometers) layer of the metallic substrate and ceramic particles, leading to enhanced mechanical interlocking [78,91].

Figure 7 demonstrates the PEEQ value over the crater surface of a substrate without a pore and with a pore of different diameters (2, 3, and 4 μm) placed at different depths (0.1$D_{Particle}$ to 0.5$D_{Particle}$) from the surface at a fixed particle diameter (15 μm) and impact velocity (500 m/s) within 24 ns of impact to examine the effect of pore diameter and depth on the PEEQ value over the Al contact surface. Figure 7 shows that the pores with the diameter of 1 and 2 μm have no clear trend on the PEEQ value over the Al surface, which is in contrast to the general trend observed for pore diameters of 3 and 4 μm. For example, a pore with $D_{Pore}$ = 2 μm at different depths slightly affects the PEEQ value compared to the PEEQ value of a substrate without a pore (see solid orange line). On the other hand, a pore with $D_{Pore}$ = 3 and 4 μm significantly impacts the PEEQ value before approximately 17.5 ns, which is denoted in the figure as t*, and defines a time for comparative purposes across all tests after which PEEQ increases linearly at more-or-less the same rate across all conditions. At $t^{*}$, the amount of PEEQ value increases, and the particle stops penetrating deeper, which is also demonstrated in Figure 5 and described in Section 3.2. This behavior can be attributed to the strain hardening that occurs at the contact-surface in the substrate at this time (between 15 and 20 ns) due to the peening effect of a hard particle on the deformable substrate, which both improves the tribological properties of the surface [92,93] and increases the PEEQ value, facilitating particle retention [78]. The impact of hard ceramic particles on a metallic matrix reduces the interfacial gaps between the matrix and particles and flattens the metallic matrix due to large plastic deformation. This helps the metallic matrix remain soft, which improves the retention of the ceramic particles (specifically the smaller size) in the matrix [94,95,96,97], increasing the deposition efficiency, and by association, the tribological and mechanical properties of the coatings [98].

### 3.6. Effect of Impact Velocity, Particle Size, and Pore Size on Time-Evolved Equivalent Plastic Strain

In this sub-section, the effect of impact velocity, particle size, pore size (at a fixed depth of 0.4$D_{Particle}$), and particle-to-pore-size ratio on the PEEQ value over the contact surface in the substrate will be further explored to quantify their effects on the PEEQ value, an indicator of particle retention [19,44,59,72,99]. Figure 8 shows the time-evolved PEEQ over the Al substrate for different simulations to investigate the effects of impact velocities (Figure 8a), particle sizes (Figure 8b), pore sizes (Figure 8c), and pore-to-particle-size ratios (Figure 8d). Figure 8a demonstrates the average PEEQ value measured over the Al contact-surface of three simulations with a fixed particle diameter of 15 μm and different impact velocities (500, 600, and 700 m/s) to examine the effect of impact velocities on an Al substrate without a pore. From Figure 8a, there is a correlation showing the higher PEEQ values for higher impact velocity. Increasing the velocity generates more kinetic energy, which results in higher plastic strain and causes a higher plastic-strain-rate over the contact surface in the substrate [100].

In Figure 8b, we present the effect of particle sizes ($D_{Particle}$ = 15, 25, and 40 μm) at a fixed impact velocity (500 m/s) on the PEEQ value to examine the particle size effect. We observe a correlation between particle diameters and PEEQ values, where the smaller particles are associated with a higher PEEQ value. The particle with $D_{Particle}$ = 15 μm has a higher PEEQ than the particle with $D_{Particle}$ = 25 μm, and $D_{Particle}$ = 25 μm has a higher PEEQ value than the particle with $D_{Particle}$ = 40 μm. While this observation may be in contrast to the fact that larger particles with larger masses result in higher kinetic energies, experimental studies in the literature [33] have shown the deposition efficiency of B${}_{4}$C with $D_{Particle}$ = 15 μm in an Al substrate is higher than the deposition efficiency corresponding to particles with $D_{Particle}$ = 25 and 40 μm. Other studies [2,86,101,102,103] have also shown non-intuitive relationships between particle size and velocity on impact deposition. While still challenging to unravel, our results show consistency between the PEEQ value of different particle sizes and the retention of the particles in experimental observations [33], indicating that determination of optimum particle size and velocity can be attributed to the PEEQ value or the plastic strain deformation over the substrate contact surface. To better understand the particle size effect on the PEEQ value, the evolution of localized plastic strain across the contact surface is explored later in Section 3.7.

Next, Figure 8c shows the PEEQ value over the contact-surface of an Al substrate containing a pore with a diameter of 1, 2, 3, and 4 μm placed at a depth of 0.4$D_{Particle}$ with a fixed particle diameter of 15 μm and impact velocities of 500, 600, and 700 m/s. By comparing Figure 8a and Figure 8c, the PEEQ value increases from 5.8 in Figure 8a to 6.0 in Figure 8c, indicating that higher PEEQ values correspond to higher impact velocities, as expected. Three more important conclusions can be drawn from Figure 8c. First, a pore of any diameter affects the PEEQ value in a nonuniform pattern at a lower impact velocity (500 m/s); a pore causes an increase then a decrease in the PEEQ value within 24 ns of deposition. Second, a pore significantly increases the PEEQ value at higher impact velocities (600 and 700 m/s). However, the relationship between the pore diameter and the increase in PEEQ value does not follow a predictable pattern. Three, comparing Figure 8c and Figure 4c, the effect of including a pore in the substrate on penetration depth (see Figure 4c,d) is greater than the effect of including a pore on the increase in PEEQ value (see Figure 8c), which might be related to the conversion of kinetic energy into penetration rather than plastic deformation and an increase in PEEQ.

Lastly, Figure 8d further explores the effect of the ratio between the pore and particle size on the PEEQ values for different particle diameters of 15, 25, and 40 μm and pore diameters of 4 and 8 μm placed at a depth of 0.4$D_{Particle}$ at a fixed impact velocity of 500 m/s. In Figure 8d, the curves associated with $\frac{D_{Pore}}{D_{Particle}}$ of 0.27, 0.16, 0.1, and 0.2 correspond to the $\frac{D_{Pore}}{D_{Particle}}=\frac{4}{15}$, $\frac{4}{25}$, $\frac{4}{40}$, and $\frac{8}{40}$, respectively. Comparing Figure 8b with Figure 8d shows that the pore-to-particle-size ratio influences PEEQ slightly. Moreover, the increase in PEEQ value for different pore-to-particle-size ratios does not follow a predictable pattern; for example, the PEEQ value for the pore-to-particle-size ratio of 0.2 ($\frac{D_{Pore}}{D_{Particle}}=\frac{8}{40}$) increases by $\frac{3.9}{2.1}$ times at 17.5 ns, which is the most significant increase in PEEQ value compared to the other cases. Overall, the comparison between Figure 8d and Figure 4d reveals that the effect of including a pore in the substrate on PEEQ is notably less than the effect of including a pore on the penetration depth, recognizing that both penetration depth and PEEQ are important for particle retention [21].

### 3.7. Effect of Pore Size, Particle Size, and Impact Velocity on the Localized Equivalent Plastic Strain in the Substrate

This final sub-section examines the effect of impact velocity, particle size, and pore size on the localized PEEQ value across the contact-surface in the substrate towards linking the effect of plastic strain localization on particle retention [78]. Here, localized PEEQ vs. normalized distance along the substrate surface is plotted to investigate the effects of impact velocity (Figure 9), particle size (Figure 10), and pore size (Figure 11). First, Figure 9 investigates the effect of impact velocity on localized PEEQ over the Al contact surface without a pore at a fixed particle diameter of 15 μm. The figure shows the time-evolved localized plastic deformation (PEEQ value) over the Al contact-surface along the distance spanning the particle diameter (2*R*) at impact velocities of 500, 600, and 700 m/s and at fixed times (i.e., t = 5, 10, 15, 20, and 24 ns). The substrate PEEQ values at times of 5 and 10 ns at the impact velocity of 500 m/s and at the time of 5 ns at an impact velocity of 600 m/s have a maximum value near the crater edges of the Al substrate (at 0.6 particle radius). This PEEQ behavior is similar to the PEEQ curve trend of the Al substrate in the Al/Al coating from the literature [44], where the PEEQ curve peaks near the edges. Figure 9 at the $V_{Impact}$ = 500 m/s also illustrates the sudden increase in PEEQ value near the crater center after 10 ns in the Al/B${}_{4}$C coating. In Figure 9, the maximum PEEQ value is observed near the crater center and also near the crater edges (0.6R) at $V_{Impact}=$ 600 and 700 m/s. Although the localized plastic strain trend is analogous at impact velocities of 600 and 700 m/s, the PEEQ value is different, and the higher PEEQ value corresponds to the higher impact velocity.

The abrupt increases in PEEQ value in Al/B${}_{4}$C observed here can be attributed to the distinct material types of the particle and substrate. When a harder ceramic particle (B${}_{4}$C) impacts a softer metallic substrate (Al), its kinetic energy transforms into plastic deformation by a cushioning mechanism, and the matrix surface acts as a cushion and is largely deformed to provide a place for the ceramic particles to retain [87]. This embedment mechanism causes localization of the plastic deformation across the crater surface [21,44,87], leading to fracture and fragmentation of the ceramic at the center [21]. The high plastic deformation at the center creates a strong bonding between particle and matrix and facilitates particle retention, as reported for the ceramic/metal coatings in the literature [79]. The discontinuous high PEEQ value at the crater edges stems from the intense pressure wave [99,104] causing jetting at the crater edges and enhancing the fragmentation and the flow of the comminuted ceramic particles [101]. Hence, we conclude that the B${}_{4}$C particle impact on the Al substrate results in a maximum PEEQ value near the crater center at all impact velocities and another maximum PEEQ value near the crater edges at higher impact velocities where the comminuted ceramic flow is more visible [79].

Next, Figure 10 examines the effect of the B${}_{4}$C particle size on the localization of the plastic strain over the contact-surface in the Al substrate by demonstrating the PEEQ value vs. the distance per particle radius (Distance/R) across the surface in the substrate for particles with diameters of 15, 25, and 40 μm at a fixed impact velocity (500 m/s) and a substrate without a pore at a range of times (5, 10, 15, 20, 24 ns). In Figure 10, an abrupt increase in PEEQ values is observed near the crater center at the impact of a particle with diameters of 15, 25, and 40 μm on a substrate. These three sub-figures show that the maximum PEEQ value near the center is higher for larger particles (the maximum magnitude corresponds to the particle with $D_{Particle}$ = 40 μm). In contrast, the PEEQ value dramatically decreases after its sudden increase, resulting in a lower average PEEQ value for larger particles (see Figure 8b), subsequently, lower particle retention, and lower deposition efficiency according to experimental data [33]. A concentration of plastic strain occurs at the center of the crater due to the high kinetic energy of impact, and the rate of deformation increases more rapidly than in the rest of the substrate, which requires more energy and stress to deform. Meanwhile, a more considerable amount of kinetic energy is released at the beginning of the particle deposition and converted into a high plastic strain, whereas there is no further stress or energy to cause another significant localized plastic strain; this is known as strain hardening [1]. Consequently, the lower average PEEQ value for larger particles can be attributed to the strain hardening phenomenon at the crater center, resulting in a considerable localized plastic strain and a significant decrease in PEEQ across the contact-surface in the substrate without an increase [1].

Finally, Figure 11 explores the effect of a pore in an Al substrate subjected to the different B${}_{4}$C impact velocities on the localized plastic deformation (PEEQ) over the contact-surface in the Al substrate. Figure 11 consists of three sub-figures showing the PEEQ value vs. the distance per particle radius (Distance/R) across the substrate surface at the time of 24 ns, when the particles stop penetrating deeper (see Figure 4c). The particle diameter is fixed ($D_{Particle}$ = 15 μm) and the particle velocities are 500, 600, and 700 m/s, and the Al substrate includes a pore with $D_{Pore}$ = 1, 2, 3, and 4 μm at a depth of 0.4$D_{Particle}$. In Figure 11, at $V_{Impact}$ = 500, 600, and 700 m/s, the PEEQ curve trend and value of the cases containing a pore with $D_{Particle}$ = 1 μm are almost identical to those of the cases without a pore, indicating the minor effect of the pore with $D_{Pore}$ = 1 μm on the PEEQ localization and magnitude, as shown in Figure 4. Additionally, there is no trend in the localized plastic strain for the cases with pore diameters of 2, 3, and 4 μm. Nevertheless, the PEEQ value tends to suddenly increase at 0.6*R* near the crater edges, as well as at 0.2*R* near the crater center, similar to Figure 9 for the substrate without a pore subjected to the impact velocities of 600 and 700 m/s. However, higher impact velocities result in higher PEEQ values in Figure 11, which has already been noted many times previously. Additionally, the sudden increase in PEEQ values near the crater center for the cases containing a pore with $D_{Pore}$ = 2, 3, and 4 μm shifts toward the peak PEEQ value near the crater edges, and this can be related to the existence of a pore at the center.

Overall, in these sub-figures, the localized PEEQ value near the crater center results in fracture and fragmentation of the B${}_{4}$C particle. The comminuted particles cause a secondary impact and the subsequent rebound of the B${}_{4}$C particles, leading to a lower deposition efficiency [101]; however, Huang et al. [12] experimentally showed that the fragmented B${}_{4}$C particles are mechanically interlocked at the crater center in the substrate due to the large plastic deformation and deeper indent of the crater, and Chakrabarty et al. [79] proved this using smoothed-particle hydrodynamics method. In addition, the particle fragmentation, spall-like processes, and the flow of the comminuted particles at the crater edges can be attributed to the maximum PEEQ value near the crater edges. Overall, our results show that the inclusion of a pore promotes damage and fracture in the ceramic particle, leading to larger plastic deformation and, subsequently, enhanced retention [21]. This brings new considerations for designing and manufacturing cold-sprayed coatings, especially those with inherent porosity and under lower-speed deposition rates [72,105].

## 4. Conclusions

In this study, the impact of a single B${}_{4}$C particle on an Al substrate in Al/B${}_{4}$C composite coatings is numerically simulated to examine the effect of impact velocity, particle size, and matrix porosity on the key particle retention parameters (i.e., penetration depth of the particle, the crater morphology, and plastic deformation (PEEQ) of the contact-surface in the substrate). The summarized key results are:Higher impact velocities, larger particles, and greater matrix porosity result in deeper penetration.Higher impact velocities and smaller particles lead to higher PEEQ values in the substrate.The effect of matrix pore size and depth on the PEEQ value is unclear.The partial or complete crush of a pore increases the non-uniform shape of the crater.A pore at low impact velocities produces a non-uniform distribution of the plastic strain and causes a complex interplay between penetration depth, contact-surface roughness, and the PEEQ value along the contact-surface in the substrate.

Overall, the results indicate that some porosity in the coating prior to deposition may improve particle retention and, by association, coating quality.

## Figures and Tables

**Figure 1 materials-16-02525-f001:**
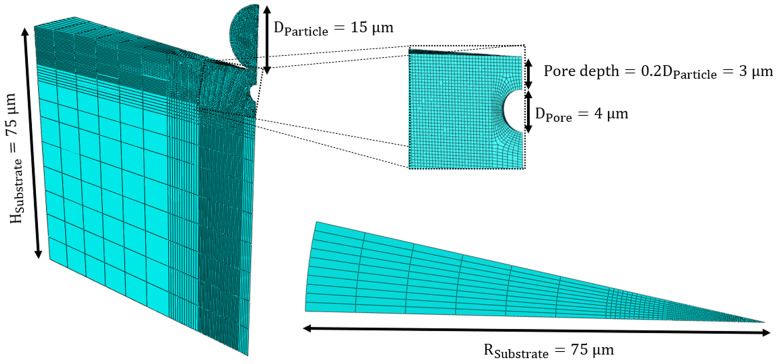
Three-dimensional numerical model geometry for simulating a single particle impact during the cold spray deposition process.

**Figure 2 materials-16-02525-f002:**
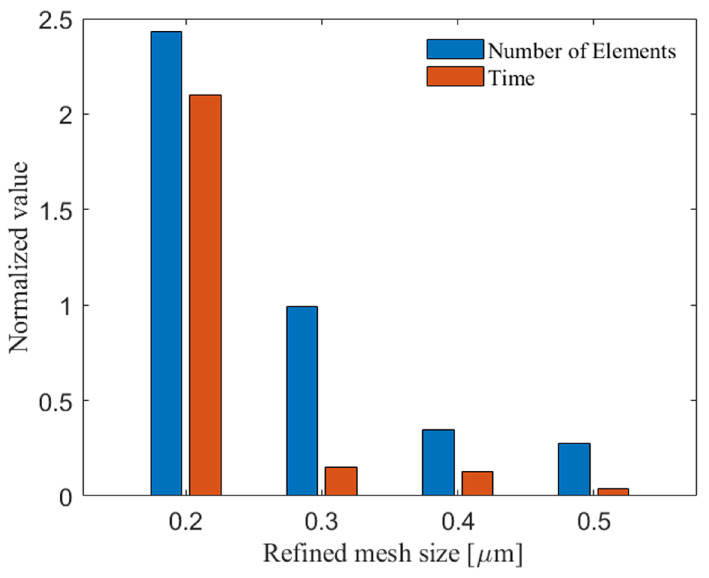
Number of elements and computational time for different refined mesh sizes (0.2, 0.3, 0.4, and 0.5 μm) using an ALE FEA framework at the particle-matrix contact areas.

**Figure 3 materials-16-02525-f003:**
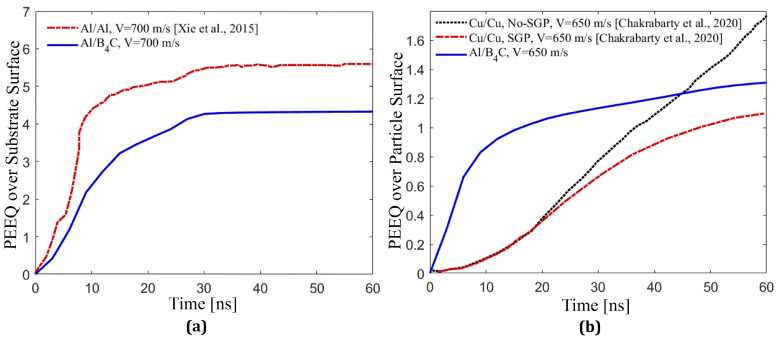
Comparison between the equivalent plastic strain (PEEQ) generated in the B${}_{4}$C particle and the Al substrate in this study and previous studies involving Al and Cu particles and Al and Cu substrates [44,45]. (**a**) The average PEEQ value over the Al contact-surface in an Al/B${}_{4}$C coating in the current study (Al/B${}_{4}$C, V = 700 m/s) and in Al/Al coating reprinted from the literature [44] (Al/Al, V = 700 m/s) with $D_{Particle}$ = 25 μm and $V_{Impact}$ = 700 m/s is calculated using an Arbitrary Lagrangian–Eulerian (ALE) and Coupled Eulerian–Lagrangian (CEL) technique, respectively, in Abaqus. The GTN material model in the current study and the original Johnson–Cook (JC) model in the literature [44] are employed. (**b**) The average PEEQ value over the entire B${}_{4}$C particle in Al/B${}_{4}$C using the JHB model (Al/B${}_{4}$C, V = 650 m/s) and ALE technique, and over the entire Cu particle in Cu/Cu coating using the modified Johnson–Cook (JC) model with and without consideration of strain gradient plasticity (Cu/Cu, SGP, V = 650 m/s and Cu/Cu, No-SGP, V = 650 m/s) and CEL technique as reprinted from reference [45]. The calculations are performed with $D_{Particle}$ = 41 μm and $V_{Impact}$ = 650 m/s.

**Figure 4 materials-16-02525-f004:**
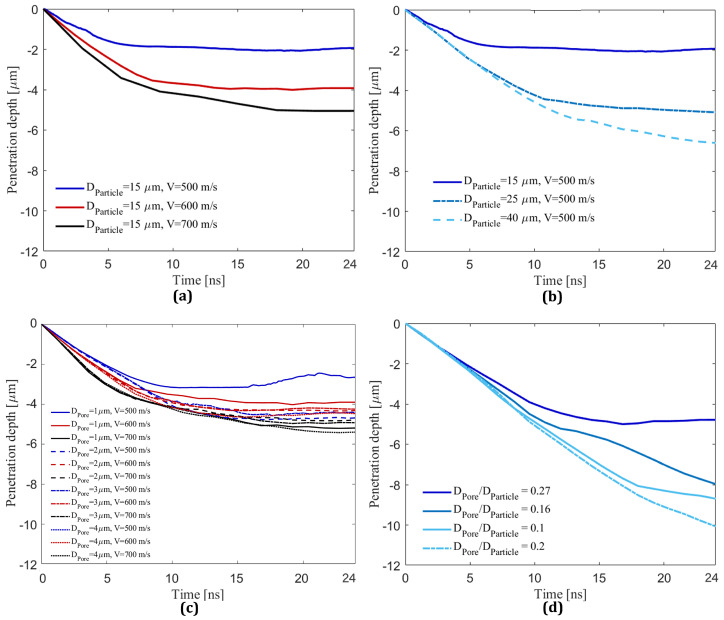
The penetration depth of the B${}_{4}$C particle in an Al substrate vs. time for varying impact velocities, particle size, and pore sizes: (**a**) The penetration depth of the particle with $D_{Particle}$ = 15 μm for $V_{Impact}$ = 500, 600, and 700 m/s. (**b**) The penetration depth of the particles with $D_{Particle}$ = 15, 25, and 40 μm and $V_{Impact}$ = 500 m/s. (**c**) The penetration depth vs. time for $D_{Particle}$ = 15 μm at $V_{Impact}$ = 500, 600, and 700 m/s on the substrate including a pore with $D_{Pore}$ = 1, 2, 3, and 4 μm placed at a depth of 0.4$D_{Particle}$. (**d**) The penetration depth of particle with $D_{Particle}$ = 15, 25, and 40 μm and $V_{Impact}$ = 500 m/s on the substrate with $D_{Pore}$ = 4 and 8 μm placed at a depth of 0.4$D_{Particle}$. The curves associated with $\frac{D_{Pore}}{D_{Particle}}$ of 0.27, 0.16, 0.1, and 0.2 correspond to $\frac{D_{Pore}}{D_{Particle}}=\frac{4}{15}$, $\frac{4}{25}$, $\frac{4}{40}$, and $\frac{8}{40}$, respectively.

**Figure 5 materials-16-02525-f005:**
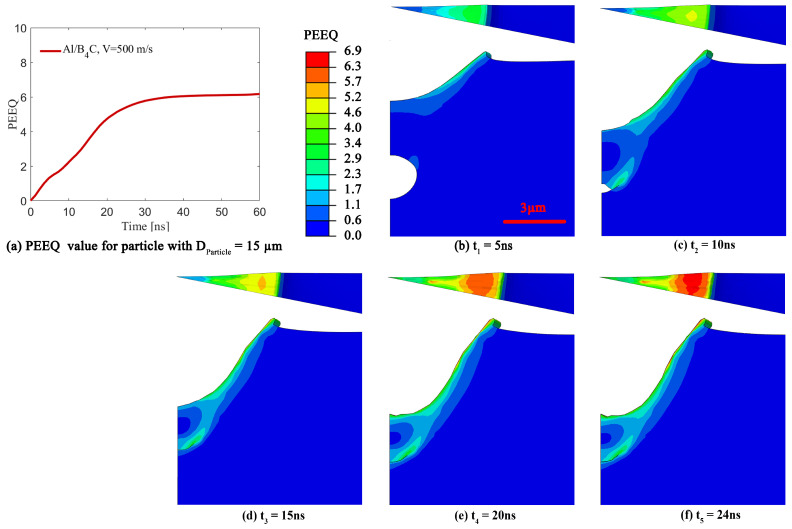
The time-resolved still frames showing pore morphology behavior and PEEQ values for a pore with $D_{Pore}$ = 4 μm at a depth of 0.3$D_{Particle}$ and $V_{Impact}$ = 500 m/s. (**a**) The average PEEQ value vs. time to determine the maximum PEEQ value which is 6. The top view and the side view of the substrate with a pore are demonstrated at a time range of (**b**) 5 ns, (**c**) 10 ns, (**d**) 15 ns, (**e**) 20 ns, and (**f**) 24 ns in order to show pore volume changes.

**Figure 6 materials-16-02525-f006:**
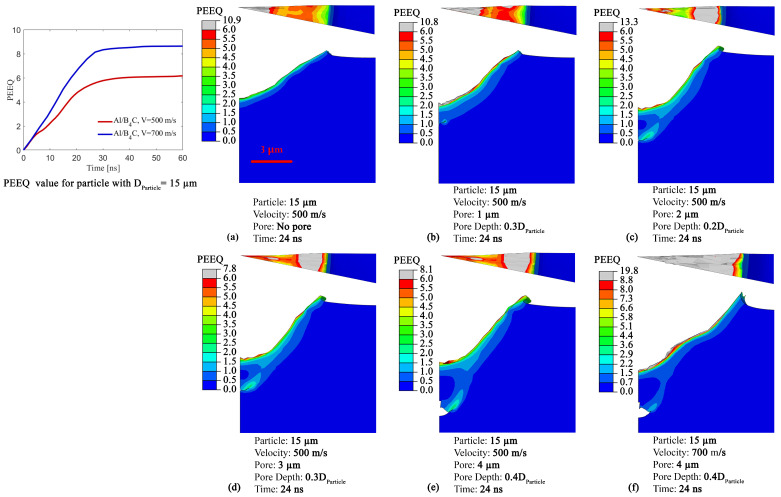
Comparison of the substrate crater morphology for $D_{Particle}$ = 15 μm cases of (**a**) Without pore and $V_{Impact}$ = 500 m/s. (**b**) With pore of $D_{Pore}$ = 1 μm at a depth of 0.3$D_{Particle}$ and $V_{Impact}$ = 500 m/s. (**c**) With pore of $D_{Pore}$ = 2 μm at a depth of 0.2$D_{Particle}$ and $V_{Impact}$ = 500 m/s. (**d**) With pore of $D_{Pore}$ = 3 μm at a depth of 0.3$D_{Particle}$ and $V_{Impact}$ = 500 m/s. (**e**) With pore of $D_{Pore}$ = 4 μm at a depth of 0.4$D_{Particle}$ and $V_{Impact}$ = 500 m/s. (**f**) With pore of $D_{Pore}$ = 4 μm at a depth of 0.4$D_{Particle}$ and $V_{Impact}$ = 700 m/s.

**Figure 7 materials-16-02525-f007:**
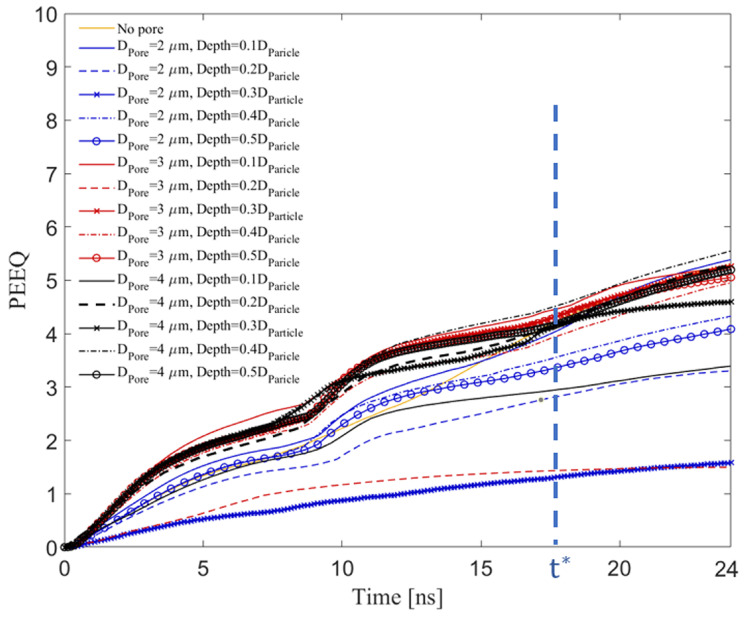
The equivalent plastic strain (PEEQ) over the contact-surface of the Al substrate and B${}_{4}$C particle for $D_{Particle}$ = 15 μm and $V_{Impact}$ = 500 m/s with a pore with diameters: $D_{Pore}$ = 2, 3, and 4 μm, placed at different depths from the surfaces (0.1$D_{Particle}$, 0.2$D_{Particle}$, 0.3$D_{Particle}$, 0.4$D_{Particle}$, and 0.5$D_{Particle}$). t* in the figure defines a time for comparative purposes across all tests after which the PEEQ increases linearly at more-or-less the same rate across all conditions.

**Figure 8 materials-16-02525-f008:**
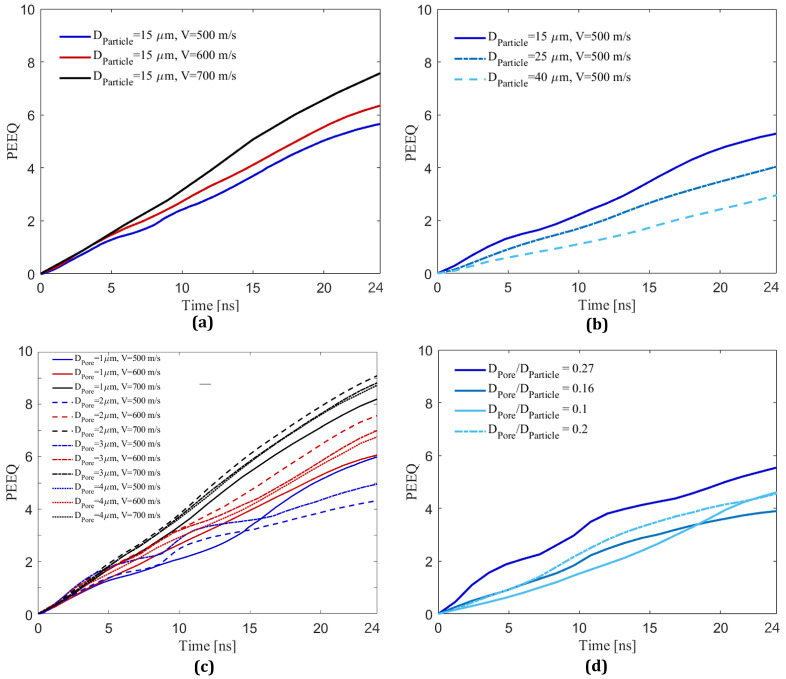
Time-evolved equivalent plastic strain (PEEQ) over the contact-surface between the Al substrate and the B${}_{4}$C particle in the Al substrate. (**a**) Particles with $D_{Particle}$ = 15 μm and $V_{Impact}$ = 500, 600, and 700 m/s. (**b**) Particles with $D_{Particle}$ = 15, 25, 40 μm and the $V_{Impact}$ = 500 m/s. (**c**) $D_{Particle}$ = 15 μm and $V_{Impact}$ = 500, 600, and 700 m/s with a pore of $D_{Pore}$ = 1, 2, 3, and 4 μm placed at a depth of 0.4$D_{Particle}$. (**d**) Particles with $D_{Particle}$ = 15, 25, and 40 μm and $V_{Impact}$ = 500 m/s impacting on a substrate including a pore with diameters of $D_{Pore}$ = 4 and 8 μm placed at a depth of 0.4$D_{Particle}$. The curves associated with $\frac{D_{Pore}}{D_{Particle}}$ of 0.27, 0.16, 0.1, and 0.2 correspond to $\frac{D_{Pore}}{D_{Particle}}=\frac{4}{15}$, $\frac{4}{25}$, $\frac{4}{40}$, and $\frac{8}{40}$, respectively.

**Figure 9 materials-16-02525-f009:**
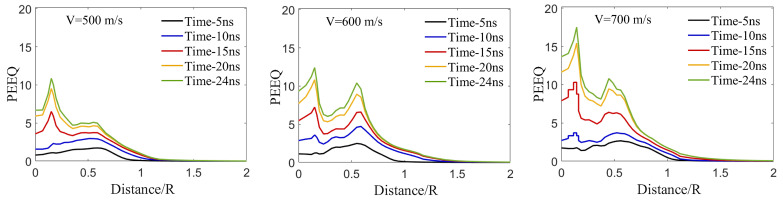
The time-evolved localized plastic deformation (PEEQ) over the contact-surface between the Al substrate and the B${}_{4}$C particle in the Al surface. The particle size is $D_{Particle}$ =15 μm and impact velocities of 500, 600, and 700 m/s on the Al substrate without pores.

**Figure 10 materials-16-02525-f010:**
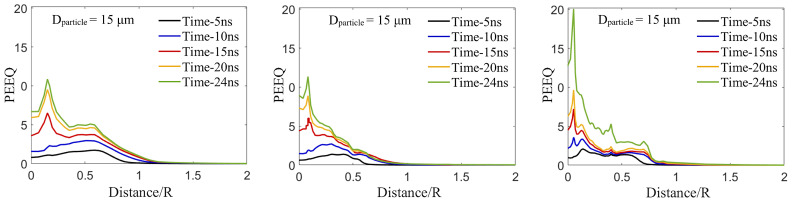
The time-evolved localized plastic deformation (PEEQ) over the contact-surface between the Al substrate and the B${}_{4}$C particle in the Al substrate for different particle diameters of 15, 25, and 40 μm at a fixed impact velocity (500 m/s) on the Al substrate without a pore.

**Figure 11 materials-16-02525-f011:**
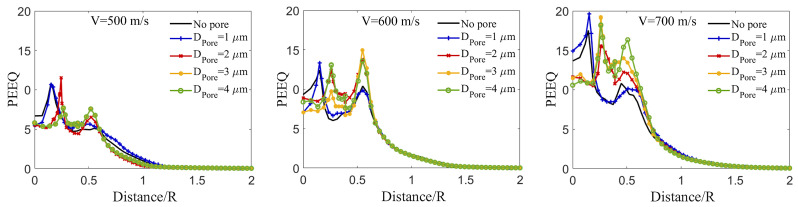
The time-evolved localized plastic deformation (PEEQ) over the contact-surface between the Al substrate and the B${}_{4}$C particle with $D_{Particle}$ = 15 μm in the Al substrate with a pore ($D_{Pore}$ = 1, 2, 3, and 4 μm) at a depth of 0.4$D_{Particle}$ at the impact velocities of 500 m/s, 600 m/s, and 700 m/s. These results are taken at 24 ns after impact for comparative purposes, with 24 ns being a time where the particle with different impact velocities no longer continues to penetrate the substrate according to Figure 4c.

**Table 1 materials-16-02525-t001:** The parameters of the JHB constitutive model used for the B${}_{4}$C particle [61].

Density	$\rho_{0}$ (kg/m${}^{3}$) = 2508	Elastic Constants	
**Damage constants**		Modulus of Elasticity	*E* (GPa) = 442
Damage coefficient	$D_{1}$ = 0.005	Poisson’s ratio	ν = 0.162
Damage exponent	*n* = 1.0	Bulk modulus	*K* (GPa) = 218
Max failure strain	$\varepsilon_{max}^{f}$ = 999.0	Shear modulus	*G* (GPa) = 190
**Strength constants**		**Pressure constants **	
Hugoniot elastic limit	HEL (GPa) = 0.27	Bulk modulus (phase 1)	$K_{1}$ (GPa) = 218
HEL strength	$\sigma_{HEL}$ (GPa) = 12.29	Pressure coefficient (phase 1)	$K_{2}$ (GPa) = 580
HEL pressure	$P_{HEL}$ (GPa) = 7.95	Pressure coefficient (phase 1)	$K_{3}$ (GPa) = 0
HEL volumetric strain	$\mu_{HEL}$ = 0.0335	Pressure coefficient (phase 2)	${\bar{K}}_{1}$ (GPa) = 307
Hydrostatic tensile strength	*T* (GPa) = 0.27	Pressure coefficient (phase 2)	${\bar{K}}_{2}$ (GPa) = 41
Intact strength constant	$\sigma_{i}$ (GPa) = 5.9	Pressure coefficient (phase 2)	${\bar{K}}_{3}$ (GPa) = 0
Intact strength constant	$P_{i}$ (GPa) = 5.9	Transition Pressure	$P_{1}$ (GPa) = 25
Max intact strength	$\sigma_{max}$ (GPa) = 12.5	Transition strain (from $P_{1}$)	$\mu_{1}$ = 0.092
Strain rate constant	*C* = 0.01	Transition pressure	$P_{2}$ (GPa) = 45
Failure strength constant	$\sigma_{f}$ (GPa) = 4.7	Transition strain (from $P_{2}$)	$\mu_{2}$ = 0.174
Failure strength constant	$P_{f}$ (GPa) = 30.0	Reference strain (phase 2)	$\mu_{0}$ = 0.03

**Table 2 materials-16-02525-t002:** The parameters of the GTN model used for the Al substrate [54,55,56].

$q_{1}$	$q_{2}$	$q_{3}$	$f_{0}$	$f_{c}$	$f_{F}$	$f_{N}$	$\varepsilon_{N}$	$S_{N}$
1.5	1	2.25	0.0017	0.02	0.0363	0.0242	0.1	0.1

**Table 3 materials-16-02525-t003:** The parameters used for numerical models of the impact deposition simulations in the Abaqus software.

Dimension Parameters		Modeling Parameters	
Particle diameter ($D_{Particle}$)	15, 25, 40 μm [33]	Analysis framework	Abaqus/explicit
Substrate size	75 μm	FEM technique	Arbitrary Lagrangian Eulerian (ALE)
Pore diameter ($D_{Pore}$)	1, 2, 3, 4 μm [2,11,12,28,33,70]	Interactions	General contact
Depth of pore	0.1 to 0.5$D_{Particle}$ [2,11,12,28,33,70]	Friction coefficient	0.25 [71]
Time	24 ns [77]	Element type	C3D8R: An 8-node linear brick [44,45,46]
Impact velocity ($V_{Impact}$)	500, 600, 700 m/s	Mesh design	Reduced integration, hourglass control
		FS	1.5 [76]

## Data Availability

The data are provided within the article.

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
