# Peer review of "Impact Deposition Behavior of Al/B_4_C Cold-Sprayed Composite Coatings: Understanding the Role of Porosity on Particle Retention"

_materials, 2023, doi:10.3390/ma16062525_

Round 1

Reviewer 1 Report

It is a very good scientific paper, well written and well argued. Very enlightening for the reader!

Author Response

Point 1: It is a very good scientific paper, well written and well argued. Very enlightening for the reader!

Response 1: The authors thank the reviewer for this comment. 

Reviewer 2 Report

The authors present a relevant study of Impact deposition behavior of Al/B4C cold-sprayed composite coatings.

I consider that the manuscript could be accepted if the authors carry out minor corrections.

The conclusions section should be improved. The authors could present their conclusions in the form of bullets to highlight each of the results obtained from their investigation.

Author Response

Point 1: The conclusions section should be improved. The authors could present their conclusions in the form of bullets to highlight each of the results obtained from their investigation.

Response 1: The authors thank the reviewer’s comment. The Conclusion section has been changed as the following:

In this study, the impact of a single B4C particle on an Al substrate in Al/B4C composite coatings is numerically simulated to examine the effect of impact velocity, particle size, and matrix porosity on the key particle retention parameters (i.e., penetration depth of the particle, the crater morphology, and plastic deformation (PEEQ) of the contact-surface in the substrate). The summarized key results are:

  • Higher impact velocities, larger particles, and greater matrix porosity result in deeper penetration.
  • Higher impact velocities and smaller particles lead to higher PEEQ values in the substrate.
  • The effect of matrix pore size and depth on the PEEQ value is unclear.
  • The partial or complete crush of a pore increases the non-uniform shape of the crater.
  • A pore at low impact velocities produces a non-uniform distribution of the plastic strain and causes a complex interplay between penetration depth, contact-surface roughness, and the PEEQ value along the contact-surface in the substrate.

Overall, the results indicate that some porosity in the coating prior to deposition may improve particle retention and, by association, coating quality.

Reviewer 3 Report

The manuscript investigates the impact deposition behavior of Al/B4C cold-sprayed composite coatings. The authors reported an exhaustive introduction, and the aim of the work are clearly stated. The methodology is properly explained and the presentation of the result well organized. For these reasons, in my opinion the work can be accepted after a minor revision.

I would make a more organized “conclusions section” with a list of brief points which synthesize the achievement of each paragraph in the “Results and Discussions section”.

Author Response

Point 1: The authors thank the reviewers for taking the time to review our manuscript and for their constructive feedback. All modifications to the manuscript that reflect the incorporation of the reviewer’s comments are included as red text in the revised draft, as well as are included in this response for ease of review. 

I would make a more organized “conclusions section” with a list of brief points which synthesize the achievement of each paragraph in the “Results and Discussions section”.

Response 1: The authors thank the reviewer’s comment. The Conclusion section has been changed as the following:

In this study, the impact of a single B4C particle on an Al substrate in Al/B4C composite coatings is numerically simulated to examine the effect of impact velocity, particle size, and matrix porosity on the key particle retention parameters (i.e., penetration depth of the particle, the crater morphology, and plastic deformation (PEEQ) of the contact-surface in the substrate). The summarized key results are:

  • Higher impact velocities, larger particles, and greater matrix porosity result in deeper penetration.
  • Higher impact velocities and smaller particles lead to higher PEEQ values in the substrate.
  • The effect of matrix pore size and depth on the PEEQ value is unclear.
  • The partial or complete crush of a pore increases the non-uniform shape of the crater.
  • A pore at low impact velocities produces a non-uniform distribution of the plastic strain and causes a complex interplay between penetration depth, contact-surface roughness, and the PEEQ value along the contact-surface in the substrate.

Overall, the results indicate that some porosity in the coating prior to deposition may improve particle retention and, by association, coating quality.

Reviewer 4 Report

Reviewer’s Comments:

The manuscript “Impact deposition behavior of Al/B4C cold-sprayed composite coatings” is a very interesting work. This study explores the role of porosity on the impact deposition of a ceramic-reinforced metal-matrix composite coating fabricated via cold spraying (i.e., Al/B4C). The Johnson-Holmquist-Beissel constitutive law and the modified Gurson-Tvergaard-Needleman model were used to describe the high strain-rate behavior of the boron carbide and the aluminum metal matrix during impact deposition, respectively. Within a finite element model framework, the Arbitrary Lagrangian- Eulerian technique is implemented to explore the roles of reinforcement particle size and velocity, and pore size and depth on particle retention by examining the post-impact crater morphology, penetration depth, and localized plastic deformation of the aluminum substrate. While I believe this topic is of great interest to our readers, I think it needs major revision before it is ready for publication. So, I recommend this manuscript for publication with major revisions.

1. In this manuscript, the authors did not explain the importance of the cold-sprayed composite coatings in the introduction part. The authors should explain the importance of cold-sprayed composite coatings.

2) Title: The title of the manuscript is not impressive. It should be modified or rewritten it.

3) Correct the following statement “The results reveal that some degree of matrix porosity may lead to improvements in particle retention. Specifically, observations made here indicate that porosity near the surface facilitates particle retention at lower velocities, while the kinetic energy dominates the particle retention at higher deposition velocities. Altogether, these results provide into deposition conditions (e.g., particle size and velocity, pore depth and size) that lead to improve particle retention and coating quality”.

4) Keywords: The cold-sprayed composite coatings are missing in the keywords. So, modify the keywords.

5) Introduction part is not impressive. The references cited are very old. So, Improve it with some latest literature like 10.1016/j.inoche.2022.109575, 10.1016/j.fuel.2021.122926

6) The authors should explain the following statement with recent references, “For employing the ALE method, the frequency is set at ten as a default value, and the number of remeshing sweeps per increment is set at 5 to 8 for various models in order to avoid errors in analysis”.

7) Add space between magnitude and unit. For example, in synthesis “21.96g” should be 21.96 g. Make the corrections throughout the manuscript regarding values and units.

8) The author should provide reason about this statement “It is observed from Figure 2 that at the mesh size of 0.3 µm, the number of elements and the associated computational time are significantly smaller than mesh sizes of 0.2 µm”.

9. Comparison of the present results with other similar findings in the literature should be discussed in more detail. This is necessary in order to place this work together with other work in the field and to give more credibility to the present results.

10) Conclusion part is very long. Make it brief and improve by adding the results of your studies.

11) There are many grammatic mistakes. Improve the English grammar of the manuscript.

Author Response

The authors thank the reviewers for taking the time to review our manuscript and for their constructive feedback. All modifications to the manuscript that reflect the incorporation of the reviewer’s comments are included as red text in the revised draft, as well as are included in this response for ease of review.

Point 1: In this manuscript, the authors did not explain the importance of the cold-sprayed composite coatings in the introduction part. The authors should explain the importance of cold-sprayed composite coatings.

The authors thank the reviewer’s comment. The importance of the cold-sprayed composite coatings is explained and the following explanation in the manuscript have been added:

Page 1,  Line 30: These features make the cold spray method a unique technique for manufacturing composite coatings, in particular, adding the other  in order to reduce material consumption and tailor the physical, mechanical, and tribological properties by blending dissimilar materials to provide commercial products for a variety of industrial applications including [21] to repair magnesium parts in aerospace [22], manufacture electro- or thermo-conductive coatings for power electronic circuit boards [23], design the orthopedic devices [22], and coat the biomedical implants [24]. Moreover, cold-sprayed composite coatings are highly appealing because they do not undergo alloying, phase transformation, or thermite reactions during fabrication [25].

Added references: [22] Cavaliere, P.; Cavaliere, L.; Lekhwani. Cold-spray coatings; Springer, 2018.

 [23] Wielage, B.; Grund, T.; Rupprecht, C.; Kuemmel, S. New method for producing power electronic circuit boards by cold-gas spraying and investigation of adhesion mechanisms. Surface and Coatings Technology 2010, 205, 1115–1118.

[24] Bandar, A.M.; Mongrain, R.; Irissou, E.; Yue, S. Improving the strength and corrosion resistance of 316L stainless steel for biomedical applications using cold spray. Surface and Coatings Technology 2013, 216, 297–307. 728

[25] Moridi, A.; Hassani-Gangaraj, S.M.; Guagliano, M.; Dao, M. Cold spray coating: review of material systems and future perspectives. Surface Engineering 2014, 30, 369–395.

Point 2: Title: The title of the manuscript is not impressive. It should be modified or rewritten it.

The authors thank the reviewer’s comment. The title of the paper is modified to the following:

Impact deposition behavior of Al/B4C cold-sprayed composite coatings: Understanding the role of porosity on particle retention

Point 3: Correct the following statement “The results reveal that some degree of matrix porosity may lead to improvements in particle retention. Specifically, observations made here indicate that porosity near the surface facilitates particle retention at lower velocities, while the kinetic energy dominates the particle retention at higher deposition velocities. Altogether, these results provide into deposition conditions (e.g., particle size and velocity, pore depth and size) that lead to improve particle retention and coating quality”.

The authors thank the reviewer for this comment. The statement in the manuscript has been corrected to:

Page 1, Line 8: Results reveal that some degree of matrix porosity may improve particle retention. In particular, porosity near the surface facilitates particle retention at lower impact velocities, while kinetic energy dominates particle retention at higher deposition velocities. Altogether, these results provide insights into the effect of deposition variables (i.e., particle size, impact velocity, pore size, and pore depth) on particle retention that improves coating quality.

Point 4: Keywords: The cold-sprayed composite coatings are missing in the keywords. So, modify the keywords.

The authors thank the reviewer for this comment. The keyword is added:

Page 1, Line 14: Keywords: Aluminum matrix; Boron carbide; Cold Spray; Cold-sprayed composite coatings; Particle reinforced metal matrix composites; Porosity

Point 5: Introduction part is not impressive. The references cited are very old. So, Improve it with some latest literature like 10.1016/j.inoche.2022.109575, 10.1016/j.fuel.2021.122926

The authors thank the reviewer for this comment. The introduction is improved by adding the papers related to the nanocomposite coatings for fuel storages [22,23]

Page 1, Line 16: Particle-reinforced metal matrix composite (PRMMC) coatings (e.g., Al/B4C [1], Al/SiC [2], Al/Al2O3 [3]) have been widely employed in a variety of applications (e.g., aerospace [4], automotive [1,5], fuel storage [6-8], and transportation [9]) because of their favorable tribological properties [6,10–12], high hardness and stiffness [10], and fatigue resistance [13]. Typical manufacturing methods for PRMMCs include friction stir [12], squeeze casting [13], stir casting [14], powder compaction [15], and thermal spraying [16]. Among these techniques, the cold spray method [17] was recently adopted because of its favorable attributes:

Added references:

[7] Kuang, C.; Tan, P.; Javed, M.; Khushi, H.H.; Nadeem, S.; Iqbal, S.; Alshammari, F.H.; Alqahtani, M.D.; Alsaab, H.O.; Awwad, N.S.; et al. Boosting Photocatalytic interaction of Sulphur doped reduced graphene oxide-based S@ rGO/NiS2 nanocomposite for destruction of pathogens and organic pollutant degradation caused by visible light. Inorganic Chemistry Communications 2022, 141, 109575.

[8] Javed, S.M.; Ahmad, Z.; Ahmed, S.; Iqbal, S.; Naqvi, I.J.; Usman, M.; Ashiq, M.N.; Elnaggar, A.Y.; El-Bahy, Z.M.; et al. Highly dispersed active sites of Ni nanoparticles onto hierarchical reduced graphene oxide architecture towards efficient water oxidation. Fuel 2022, 312, 122926.

Point 6: The authors should explain the following statement with recent references, “For employing the ALE method, the frequency is set at ten as a default value, and the number of remeshing sweeps per increment is set at 5 to 8 for various models in order to avoid errors in analysis”.

The authors thank the reviewer’s comment. The statement is cited using the recent related references [65, 68, 69, 70]

Page 7, Line 219: For employing the ALE method [70], the frequency is set at ten as a default value [70,73], and the number of remeshing sweeps per increment is set between 5 to 8 for various models in order to avoid errors in analysis [74 ,75].

Added references: [70] Lordejani, A.A.; Vitali, L.; Guagliano, M.; Bagherifard, S. Estimating deposition efficiency and chemical composition variation along thickness for cold spraying of composite feedstocks. Surface and Coatings Technology 2022, 436, 128239.

[73] Delloro, F.; Jeandin, M.; Jeulin, D.; Proudhon, H.; Faessel, M.; Bianchi, L.; Meillot, E.; Helfen, L. A morphological approach to the modeling of the cold spray process. Journal of Thermal Spray Technology 2017, 26, 1838–1850.
[74] Rahmati, S.; Jodoin, B. Physically based finite element modeling method to predict metallic bonding in cold spray. Journal of Thermal Spray Technology 2020, 29, 611–629.
[75] Zhu, L.; Jen, T.C.; Pan, Y.T.; Chen, H.S. Particle bonding mechanism in cold gas dynamic spray: a three-dimensional approach. Journal of Thermal Spray Technology 2017, 26, 1859–1873.

Point 7: Add space between magnitude and unit. For example, in synthesis “21.96g” should be 21.96 g. Make the corrections throughout the manuscript regarding values and units.

The authors thank the reviewer for this comment. The Tables and the texts are reviewed, and a space added between the magnitude and unit.

Point 8: The author should provide reason about this statement “It is observed from Figure 2 that at the mesh size of 0.3 µm, the number of elements and the associated computational time are significantly smaller than mesh sizes of 0.2 µm”.

The authors thank the reviewer’s comment. The statement in the manuscript has been changed to:

Page 9, Line 280: It is observed from Figure 2 that at the mesh size of 0.3 µm (with the normalized computational time of 0.148 and the normalized number of elements of 0.991), the number of elements and the associated computational time are notably lower than mesh sizes of 0.2 µm (with the normalized computational time of 2.163 and the normalized number of elements of 2.431).

Point 9: Comparison of the present results with other similar findings in the literature should be discussed in more detail. This is necessary in order to place this work together with other work in the field and to give more credibility to the present results.

The authors thank the reviewer for this comment. Explanations have been added in the manuscript:

Page 11, Line 340: This is well aligned with the numerical results for different impact velocities in the literature [43] since the higher impact velocities result in greater kinetic energy that is the main driver of particle retention in the cold spray technique [19].

Page 13, Line 423: High impact velocities promote the plastic deformation of the matrix, resulting in surface hardening through the tamping effect that leads to collapse of the interfacial gaps, flaws, and surface porosity, as well as strengthens the bonding at the metal/ceramic interfaces [21,89].  

Page 16, Line 492: The impact of hard ceramic particles on a metallic matrix reduces the interfacial gaps between matrix and particles and flattens the metallic matrix as a result of large plastic deformation. This helps the metallic matrix remain soft which improves the retention of the ceramic particles (specifically the smaller size) in the soft matrix [94-97], increasing the deposition efficiency, and by association, the tribological and mechanical properties of the coatings [98].

Page 17, Line 522: While still challenging to unravel, our results show a consistency between the PEEQ value of different particle sizes and the retention of the particles in experimental observations [31], indicating that determination of optimum particle size and velocity can be attributed to the
PEEQ value or the plastic strain deformation over the substrate contact surface.

Page 19, Line 580:  When a harder ceramic particle (B4C) impacts a softer metallic substrate (Al), its kinetic energy transforms into plastic deformation by a cushioning mechanism, and the matrix surface acts as a cushion and is largely deformed to provide a bed for the ceramic particles to retain [87]. This embedment mechanism causes localization of the plastic deformation across the crater surface [21,44,87], leading to fracture and fragmentation of the ceramic at the center [21].

Added references:  [93] Imbriglio, S.I.; Chromik, R.R. Factors affecting adhesion in metal/ceramic interfaces created by cold spray. Journal of Thermal Spray Technology 2021, pp. 1–21.
[94] Yu, M.; Saito, H.; Bernard, C.; Ichikawa, Y.; Ogawa, K. Influence of the Low-Pressure Cold Spray Operation Parameters on Coating Properties in Metallization of Ceramic Substrates Using Copper and Aluminum Composite Powder. In Proceedings of the ITSC2021. ASM International, 2021, pp. 147–152.
[95] Yu, M.; Ichikawa, Y.; Ogawa, K. Development of Cu Coating on Ceramic Substrates by Low Pressure Cold Spray and its Deposition Mechanism Analysis. In Proceedings of the Materials Science Forum. Trans Tech Publ, 2021, Vol. 1016, pp. 1703–1709.
[96] Tregenza, O.; Saha, M.; Hutasoit, N.; Hulston, C.; Palanisamy, S. An experimental evaluation of the thermal interface resistance between cold sprayed copper/laser-textured alumina bi-layered composites. International Journal of Heat and Mass Transfer 2022, 188, 122606.

Point 10: Conclusion part is very long. Make it brief and improve by adding the results of your studies.

The authors appreciate reviewer’s comment. The Conclusion section has been changed as the following:

In this study, the impact of a single B4C particle on an Al substrate in Al/B4C composite coatings is numerically simulated to examine the effect of impact velocity, particle size, and matrix porosity on the key particle retention parameters (i.e., penetration depth of the particle, the crater morphology, and plastic deformation (PEEQ) of the contact-surface in the substrate). The summarized key results are:

  • Higher impact velocities, larger particles, and greater matrix porosity result in deeper penetration.
  • Higher impact velocities and smaller particles lead to higher PEEQ values in the substrate.
  • The effect of matrix pore size and depth on the PEEQ value is unclear.
  • The partial or complete crush of a pore increases the non-uniform shape of the crater.
  • A pore at low impact velocities produces a non-uniform distribution of the plastic strain and causes a complex interplay between penetration depth, contact-surface roughness, and the PEEQ value along the contact-surface in the substrate.

Overall, the results indicate that some porosity in the coating prior to deposition may improve particle retention and, by association, coating quality.

Point 11: There are many grammatic mistakes. Improve the English grammar of the manuscript.

The authors appreciate reviewer’s comment. The manuscript is revised, and the grammatical mistakes are amended where needed.

Reviewer 5 Report

The present work entitled "Impact deposition behavior of Al/B4C cold-sprayed composite coatings" considers the study of the porosity influence on the process of cold spraying.

The authors conducted a good literary review in the article. The methodological part is described in detail.

Large number of computer models were calculated and all the results are described in detail.

Nevertheless, there are several comments to the authors:

It is not clear why the authors chose the sphere as a model of a boron particle. It is known that particles of carbide boron usually have a faceted shape due to the fragility of the material.

What influence does the form of particles have on the calculation results?

A similar question for the form of pores.

Descriptions to figures are very cumbersome and difficult to read. A lot of reasoning in the description of the figures. It must be written in the text of the article. Excessive reasoning in the description of the figures is highlighted in yellow. (check the file)

Refs in conclusion. Only the conclusions obtained in this study must be given, and not refs to other studies.

It is recommended to accept the article for publication after minor revision.

Author Response

The authors thank the reviewers for taking the time to review our manuscript and for their constructive feedback. All modifications to the manuscript that reflect the incorporation of the reviewer’s comments are included as red text in the revised draft for ease of review.

Point 1: It is not clear why the authors chose the sphere as a model of a boron particle. It is known that particles of carbide boron usually have a faceted shape due to the fragility of the material.

What influence does the form of particles have on the calculation results?

A similar question for the form of pores.

Response1: The authors thank the reviewer for this comment. As the manuscript aims to examine how pores affect particle retention parameters, spherical shapes of particles and pores were considered in order to simplify simulations. To the best of the authors' knowledge, the effect of pore size and shape on particle retention has not been studied yet, and this could represent a future study by the authors.

Based on the reviewer's comments, the following statements have been added in the manuscript to explain the reasons

Page 6, Line 189: While the B4C particle has an irregular morphology [65], which results in better retention in the Al matrix and higher reinforcement contents, in this study, the particle shape is assumed to be spherical to simplify the simulation. Ceramic particles with spherical shape may have a lower reinforcement content [66], but they are more likely to increase the in-situ hammering effect, which enhances grain refinement and structure density [67].

Added references: [65] Nicewicz, P.; Peciar, P.; Macho, O.; Sano, T.; Hogan, J.D. Quasi-static confined uniaxial compaction of granular alumina and boron carbide observing the particle size effects. Journal of the American Ceramic Society 2020, 103, 2193–2209.

[66] Qiu, X.; Wang, J.q.; Tang, J.r.; Gyansah, L.; Zhao, Z.p.; Xiong, T.y.; et al. Microstructure, microhardness and tribological behavior of Al2O3 reinforced A380 aluminum alloy composite coatings prepared by cold spray technique. Surface and Coatings Technology 2018, 350, 391–400.
[67] Xie, X.; Yin, S.; Raoelison, R.n.; Chen, C.; Verdy, C.; Li, W.; Ji, G.; Ren, Z.; Liao, H. Al matrix composites fabricated by solid-state cold spray deposition: A critical review. Journal of Materials Science & Technology 2021, 86, 20–55.

Page 7, Line 209: In addition, the pore shape is assumed to be spherical to simplify the simulation, and the pore diameters are selected to be 1, 2, 3, and 4 µm and pore depths of 0.1DParticle-0.5DParticle based on the observation made in microscopic images of Al/B4C composites from the literature [2, 9,10 ,26 ,31 ,65].

Point 2: Descriptions to figures are very cumbersome and difficult to read. A lot of reasoning in the description of the figures. It must be written in the text of the article. Excessive reasoning in the description of the figures is highlighted in yellow. (check the file)

Response 2: The authors thank the reviewer’s comment. The excessive description highlighted in yellow in the figures’ captions are eliminated and following captions are:

Figure 1. Three-dimensional numerical model geometry for simulating a single particle impact during the cold spray deposition process.

Figure 2. Number of elements and computational time for different refined mesh sizes (0.2, 0.3, 0.4, and 0.5 µm) using an ALE FEA framework at the particle-matrix contact areas.

Figure 3. Comparison between the equivalent plastic strain (PEEQ) generated in the B4C particle
and the Al substrate in this study and previous studies involving Al and Cu particles and Al and Cu substrates [40,41]. (a) The average PEEQ value over the Al contact-surface in an Al/B4C coating (the current study) and in Al/Al coating (reprinted from literature [40]) with DParticle= 25 µm and VImpact= 700 m/s is calculated using Arbitrary Lagrangian-Eulerian (ALE) and Coupled Eulerian-Lagrangian (CEL) technique, respectively, in Abaqus. The GTN material model in the current study and the original Johnson-Cook (JC) model in the literature [40] are employed. (b) The average PEEQ value over the entire B4C particle in Al/B4C using the JHB model and Cu in Cu/Cu coating using the modified Johnson-Cook (JC) model with and without considering strain gradient plasticity (SGP and No-SGP) reprinted from reference [41] with DParticle= 41 µm and VImpact= 650 m/s is obtained using ALE and CEL techniques, respectively.

Figure 4. The penetration depth of the B4C particle in an Al substrate vs. time for varying impact velocities, particle size, and pore sizes: (a) The penetration depth of the particle with DParticle= 15 µm for VImpact= 500, 600, and 700 m/s. (b) The penetration depth of the particles with DParticle= 15, 25, and 40 µm and VImpact=500 m/s. (c) The penetration depth vs. time for DParticle= 15 µm at VImpact= 500, 600, and 700 m/s on the substrate including a pore with DPore= 1, 2, 3, and 4 µm placed at a depth of 0.4DParticle. (d) The penetration depth of particle with DParticle= 15, 25, and 40 µm and VImpact= 500 m/s on the substrate with DPore= 4 and 8 µm placed at a depth of 0.4DParticle. The curves associated with DPore /DParticle of 0.27, 0.16, 0.1, and 0.2 correspond to DPore /DParticle = 4/15 , 4/25 , 4/40 , and 8/40, respectively.

Figure 5. The time-resolved still frames showing pore morphology behavior and PEEQ values for a pore with DPore= 4 µm at a depth of 0.3DParticle and VImpact= 500 m/s. (a) The average PEEQ value vs. time to determine the maximum PEEQ value which is 6. The top view and the side view of the
substrate with a pore are demonstrated at a time range of (b) 5 ns, (c) 10 ns, (d) 15 ns, (e) 20 ns, and (f) 24 ns in order to show pore volume changes.

Figure 6. Comparison of the substrate crater morphology for DParticle= 15 cases of: (a) Without pore and VImpact= 500 m/s. (b) With pore of DPore= 1  at a depth of 0.3DParticle and VImpact= 500 m/s. (c) With pore of DPore= 2  at a depth of 0.2DParticle and VImpact= 500 m/s. (d) With pore of DPore= 3  at a depth of 0.3DParticle and VImpact= 500 m/s. (e) With pore of DPore= 4  at a depth of 0.4DParticle and VImpact= 500 m/s. (f) With pore of DPore= 4  at a depth of 0.4DParticle and VImpact= 700 m/s.

Figure 7. The equivalent plastic strain (PEEQ) over the contact-surface of the Al substrate and B4C particle for DParticle= 15  and VImpact= 500 m/s with a pore with diameters: DPore= 2, 3, and 4 , placed at different depths from the surfaces (0.1DParticle, 0.2DParticle, 0.3DParticle, 0.4DParticle, and 0.5DParticle). t* in the figure defines a time for comparative purposes across all tests after which the PEEQ increases linearly at more-or-less the same rate across all conditions.

Figure 8. Time-evolved equivalent plastic strain (PEEQ) over the contact-surface between the Al substrate and the B4C particle in the Al substrate. (a) Particles with DParticle=15  and VImpact= 500, 600, and 700 m/s. (b) Particles with DParticle= 15, 25, 40  and the VImpact= 500 m/s. (c) DParticle= 15  and VImpact= 500, 600, and 700 m/s with a pore of DPore= 1, 2, 3, and 4  placed at a depth of 0.4DParticle. (d) Particles with DParticle= 15, 25, and 40  and VImpact= 500 m/s impacting on a substrate including a pore with diameters of DPore= 4 and 8  placed at a depth of 0.4DParticle. The curves associated with DPore /DParticle of 0.27, 0.16, 0.1, and 0.2 correspond to DPore/DParticle = 4/15 , 4/25 , 4/40 , and 8/40, respectively.

Figure 9. The time-evolved localized plastic deformation (PEEQ) over contact-surface between the Al substrate and the B4C particle in the Al surface. The particle size is DParticle=15  and impact velocities of 500, 600, and 700 m/s on the Al substrate without pores.

Figure 10. The time-evolved localized plastic deformation (PEEQ) over the contact-surface between the Al substrate and the B4C particle in the Al substrate for different particle diameters of: 15, 25, and 40 µm at a fixed impact velocity (500 m/s) on the Al substrate without a pore.

Figure 11. The time-evolved localized plastic deformation (PEEQ) over the contact-surface between the Al substrate and the B4C particle with DParticle= 15 µm in the Al substrate with a pore (DPore= 1, 2, 3, and 4 µm) at a depth of 0.4DParticle at the impact velocities of 500 m/s, 600 m/s, and 700
m/s. These results are taken at 24 ns after impact for comparative purposes, with 24 ns being a time where the particle with different impact velocities no longer continues to penetrate the substrate according to Figure 4c.

Point 3: Refs in conclusion. Only the conclusions obtained in this study must be given, and not refs to other studies.

Response 3: The authors appreciate reviewer’s comment. The Conclusion section has been changed as the following:

In this study, the impact of a single B4C particle on an Al substrate in Al/B4C composite coatings is numerically simulated to examine the effect of impact velocity, particle size, and matrix porosity on the key particle retention parameters (i.e., penetration depth of the particle, the crater morphology, and plastic deformation (PEEQ) of the contact-surface in the substrate). The summarized key results are:

  • Higher impact velocities, larger particles, and greater matrix porosity result in deeper penetration.
  • Higher impact velocities and smaller particles lead to higher PEEQ values in the substrate.
  • The effect of matrix pore size and depth on the PEEQ value is unclear.
  • The partial or complete crush of a pore increases the non-uniform shape of the crater.
  • A pore at low impact velocities produces a non-uniform distribution of the plastic strain and causes a complex interplay between penetration depth, contact-surface roughness, and the PEEQ value along the contact-surface in the substrate.

Overall, the results indicate that some porosity in the coating prior to deposition may improve particle retention and, by association, coating quality.
